# Not All Low-Pass Filters are Robust
# in Graph Convolutional Networks

**Heng Chang**[1]*, **Yu Rong**[2], **Tingyang Xu**[2], **Yatao Bian**[2], **Shiji Zhou**[1],
**Xin Wang**[3]†, **Junzhou Huang**[2], and **Wenwu Zhu**[3]†

[1]Tsinghua-Berkeley Shenzhen Institute, Tsinghua University
[2]Tencent AI Lab  [3]Department of Computer Science and Technology, Tsinghua University
{changh17,zsj17}@mails.tsinghua.edu.cn, {xin_wang,wwzhu}@tsinghua.edu.cn,
yu.rong@hotmail.com, Tingyangxu@tencent.com, yatao.bian@gmail.com, jzhuang75@gmail.com

## Abstract

Graph Convolutional Networks (GCNs) are promising deep learning approaches in
learning representations for graph-structured data. Despite the proliferation of such
methods, it is well known that they are vulnerable to carefully crafted adversarial
attacks on the graph structure. In this paper, we first conduct an adversarial
vulnerability analysis based on matrix perturbation theory. We prove that the low-
frequency components of the symmetric normalized Laplacian, which is usually
used as the convolutional filter in GCNs, could be more robust against structural
perturbations when their eigenvalues fall into a certain robust interval. Our results
indicate that not all low-frequency components are robust to adversarial attacks and
provide a deeper understanding of the relationship between graph spectrum and
robustness of GCNs. Motivated by the theory, we present GCN-LFR[3], a general
robust co-training paradigm for GCN-based models, that encourages transferring
the robustness of low-frequency components with an auxiliary neural network. To
this end, GCN-LFR could enhance the robustness of various kinds of GCN-based
models against poisoning structural attacks in a plug-and-play manner. Extensive
experiments across five benchmark datasets and five GCN-based models also
confirm that GCN-LFR is resistant to the adversarial attacks without compromising
on performance in the benign situation.

## 1 Introduction

Graph Convolutional Networks (GCNs) elaborate the expressive power of deep learning from
grid-like data to graph-structured data and have achieved remarkable success in a wide variety
of domains [7, 6, 13, 27, 22, 30, 1, 8, 42, 18, 31, 12, 41, 19]. Just like CNNs, modern GCNs
could promisingly learn both the local and global structural patterns of graphs through designed
convolutions. However, the vulnerability of GCNs against adversarial attacks has been revealed
recently [70, 11, 9]. The lack of robustness arouses concerns on applying GCNs in a variety of
fields pertaining to security and privacy. Adversarial attacks on graphs aim to degrade the ability
of representation learning of GCNs, and fool them to make wrong decisions by perturbing either
node features or graph structures. Given the complexity of the underlying structural information and
the ease of operation in practice [50], the majority of literature focuses on the adversarial attacks on
structures by inserting/removing/rewiring adversarial edges, *i.e.*, structural attacks. In other words,

---

*Heng Chang is supported by 2020 Tencent Rhino-Bird Elite Training Program. Work done during Heng's
internship at Tencent AI Lab.

†Corresponding authors.

[3]Code available at https://github.com/SwiftieH/LFR.

35th Conference on Neural Information Processing Systems (NeurIPS 2021).

structural attacks on graphs during the training stage of GCNs with node features unchanged becomes a commonly considered setting by attackers [24]. Therefore, it is of emerging importance to enhance the robustness of GCNs against structural attacks, which is also the aim of this work.

For a better understanding of the structural attacks, it naturally leads to the question: *Do the adversarial edges post equal influences on the graph spectrum*? Here, graph spectrum plays a significant role in Graph Signal Processing (GSP) [45, 39] and graph learning. Empirical observations from recent papers suggest the answer could be "no". We can observe that the perturbations resulting from structural attacks express an implicit trend on the graph spectrum [14]. As the toy example shown in Figure 1, low-frequency components (far left) tend to be more robust in comparison with high-frequency ones (far right). Similarly, [29] observes that the low-pass filters seem to be more stable than high-pass filters when we randomly add/remove edges. These observations point to a promising direction for developing principled defense approaches based on the theoretical analysis of graph spectrum.

**Contributions.** To mitigate the vulnerability of GCNs against adversarial attacks, in this paper, we aim to bridge the gap between the spectral analysis and the robustness of GCNs under structural perturbations. Specifically, for the symmetric normalized Laplacian, which is commonly chosen as the graph filter in GCNs, we prove that the low-frequency components could be more robust against both one-edge and multi-edge perturbations when their eigenvalues fall into a robust interval. Interestingly, this result reveals that *not all low-frequency components are robust to adversarial attacks*.

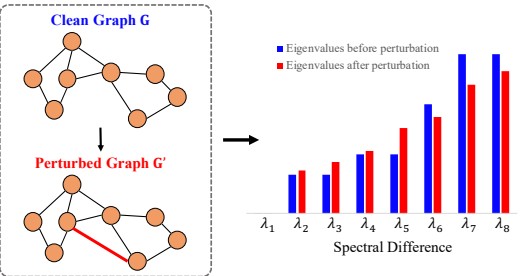

Figure 1: Observation. After adversarial attacks on the structure of graphs, the perturbations on the low-frequency components (far left) are smaller than that in the high-frequency ones (far right).

Considering the benefit from robust interval in low frequencies, we further propose GCN-LFR (**L**ow-**F**requency based **R**egularization), a general robust co-training paradigm that can transfer the robustness from the eligible low-frequency components. GCN-LFR is a general defender that can work with any GCN-based model against various training-time attacks. In particular, GCN-LFR regularizes the training process of any given GCN with the robust information from an auxiliary regularization net $\mathcal{M}_{\text{LFR}}$, which injects a set of learnable parameters to imitate the robust interval for low-frequency components. Moreover, instead of jointly training two branches, we design an alternative training scheme to accelerate the training process. We observe that the alternative training scheme also benefits the final performance in practice.

In experiments, to demonstrate the flexibility of GCN-LFR, we integrate GCN-LFR with five popular GCN-based models and compare GCN-LFR with four state-of-the-art defenders across five benchmark datasets under a variety of settings, including one-edge targeted, multi-edge targeted, and non-targeted attacks. Extensive results demonstrate that GCN-LFR consistently outperforms the defending baselines under all settings. Remarkably, we also show that GCN-LFR successfully enhances the robustness of GCNs without deduction of performance on benign graphs. It further reveals the broad applicability and relevance of GCN-LFR in the area of deep graph learning.

## 2 Related Work

**Adversarial attacks on graphs.** Adversarial attacks on graph neural networks have drawn unprecedented attention from researchers recently [47, 24, 49]. Based on the stage where the attack happens, adversarial attacks on graph-structured data can be divided into 1) *poisoning attacks* (*e.g.*, Nettack [70]) that perturb the graph in the training stage and 2) *evasion attacks* (*e.g.*, RL-S2V [11]) that perturb the graph at testing. A series of studies integrates various of techniques to construct persuasive adversarial samples, such as gradient based methods [55, 60], meta-learning [71], matrix perturbation theory [3], graph generation [32], graph signal processing [9] and reinforcement learning [11, 48].

**Defenses on graphs.** Other than the adversarial attacks on graphs, researchers are also concerned about the robustness of graph models. Generally, we can category the existing methods into two families: spatial-based and spectral-based defense. 1) *Spatial-based defense*. GCN-Jaccard [55] proposes to examine fake edges in the step of preprocessing by utilizing similarity metrics. RGCN [67],

Pro-GNN [25], and VPN [23] focus on proposing new variants of GCNs that can effectively defend against attacks on VanillaGCN. GNNGUARD [63] detects the adverse effects existing in the relationship between the graph structure and node features by neighbor importance estimation. Low-Pass [53] considers low-pass message passing from neighborhoods, which is similar to the attention mechanism in GNNGUARD to defense structural attacks. 2) *Spectral-based defense*. GCN-SVD [14] empirically verifies that only the high-rank singular components of the graph are affected by Nettack, and preprocesses the poisoned graph via SVD with pure structural information for defense. In this paper, we first explore the theoretical connection between robustness and graph spectrum, and propose GCN-LFR to bring the robustness information from eligible low-frequency components in the spectral domain. The proposed framework GCN-LFR is a general defender and can protect various GCN-based models from adversarial attacks.

## 3 Preliminaries

We consider an undirected attributed graph $\mathcal{G} = (\mathcal{V}, \mathcal{E}, \boldsymbol{X})$, where $|\mathcal{V}| = N$ is the set of $N$ nodes, $\mathcal{E} \subseteq \mathcal{V} \times \mathcal{V}$ is the set of $E$ edges, and $\boldsymbol{X} = \{\boldsymbol{x}_1, ..., \boldsymbol{x}_N\}$ is the associated feature matrix, in which $\boldsymbol{x}_u \in \mathbb{R}^M$ is the $M$-dimensional node feature for node $u \in \mathcal{V}$. We utilize $\mathcal{N}_u$ to denote the neighbors of node $u$. We denote the adjacency matrix as $\boldsymbol{A} \in \mathbb{R}^{N \times N}$, where $\boldsymbol{A}_{uv} \in \{0, 1\}$ indicates whether an edge $e_{uv} \in \mathcal{E}$ exists. The node degree matrix is defined as $\boldsymbol{D} = \text{diag}(d_1, \cdots, d_n)$, where $d_u = \sum_v \boldsymbol{A}_{uv}$ represents the degree of node $u$. An adversarial attacker deliberately perturbs edges in $\mathcal{G}$ and results in the poisoned version of $\mathcal{G}$, which we denote as $\mathcal{G}' = (\mathcal{V}', \mathcal{E}', \boldsymbol{X})$, and $\boldsymbol{A}'$ is the perturbed version of $\boldsymbol{A}$. Let $\mathcal{L} = \boldsymbol{D}^{-1/2}(\boldsymbol{D} - \boldsymbol{A})\boldsymbol{D}^{-1/2}$ be the symmetric normalized graph Laplacian. $\mathcal{L}$ has eigenvalues ranging from 0 to 2. $\hat{\boldsymbol{A}} = \boldsymbol{D}^{-\frac{1}{2}}\boldsymbol{A}\boldsymbol{D}^{-\frac{1}{2}} = \boldsymbol{I}_n - \mathcal{L}$ is the symmetric normalized adjacency matrix, whose eigenvalues range from $-1$ to $1$. Specifically, the *low-frequency components* refer to the ones of $\mathcal{L}$ with eigenvalues in $[0, 1)$ (of $\hat{\boldsymbol{A}}$ with eigenvalues in $(0, 1]$), and the *high-frequency* ones refer to that of $\mathcal{L}$ with eigenvalues in $(1, 2]$ (of $\hat{\boldsymbol{A}}$ with eigenvalues in $[-1, 0)$).

### 3.1 Graph Convolutional Networks

VanillaGCN can be viewed as the spectral convolution based on the Fourier transform on graphs with first-order Chebyshev polynomial filter $g(\mathcal{L})$. The layer-wise update rule is given as:

$$\text{feature transformation}: \boldsymbol{H}'^{(l)} = \boldsymbol{H}^{(l)}\boldsymbol{\Theta}, \quad \text{graph convolution}: \boldsymbol{H}^{(l+1)} = \sigma(g(\mathcal{L})\boldsymbol{H}'^{(l)}), \qquad (1)$$

where $\boldsymbol{H}^{(l+1)} \in \mathbb{R}^{N \times q^{(l+1)}}$ is the output from hidden layer $l$ with input as $\boldsymbol{H}^{(l)} \in \mathbb{R}^{N \times q^{(l)}}$. $q^{(l)}$ refers to the dimension of output at layer $l$. For the input layer, we have $\boldsymbol{H}^{(0)} = \boldsymbol{X}$. $\sigma(\cdot)$ refers to the activation function, such as ReLU. $\boldsymbol{\Theta}^l \in \mathbb{R}^{q^{(l)} \times q^{(l+1)}}$ refers to the weight matrix. In Eq.(1), different filter $g(\mathcal{L})$ leads to different GCNs. For example, VanillaGCN along with its variants [61, 54, 16, 10] usually use the symmetric normalized adjacency matrix $\hat{\boldsymbol{A}} = \boldsymbol{I}_n - \mathcal{L}$ as $g(\mathcal{L})$. Therefore, we mainly focus on the analysis of filter $\hat{\boldsymbol{A}}$, and try to find out the connection between the robustness of GCNs and the perturbation on $\hat{\boldsymbol{A}}$. To ease the notation, we denote $\boldsymbol{U}$ and $\boldsymbol{\Lambda} = \text{diag}(\lambda_1, \cdots, \lambda_N)$ as the eigen-pairs of $\hat{\boldsymbol{A}}$ in the following sections unless otherwise indicated.

### 3.2 Poisoning Structural Attacks

Poisoning structural attacks towards graph-structured data focus on corrupting the topology during training stage [24]. Specifically, the attacker is able to *delete* or *insert* a small number of edges on the benign graph to generate a perturbed graph $\mathcal{G}'$ with adjacency matrix $\boldsymbol{A}'$. Through this manipulation, the attacker intends to fool the prediction of GCNs and damage the performance of the downstream learning tasks. Formally, given a fixed budget of perturbation $\Delta$ on edges, *i.e.*, $||\boldsymbol{A}' - \boldsymbol{A}||_0 \le 2\Delta$, the adversarial attack on a GCN-based model $\mathcal{M}$ can be formulated as a bi-level optimization problem [3, 9]:

$$\underset{\boldsymbol{A}' \in \mathcal{P}_\Delta^\mathcal{G}}{\arg\min} \ \mathcal{L}_{\text{attack}}(\mathcal{M}(\boldsymbol{A}', \boldsymbol{X}; \boldsymbol{\Theta}^*), \boldsymbol{y}) \quad \text{s.t.} \quad \boldsymbol{\Theta}^* = \underset{\boldsymbol{\Theta}}{\arg\min} \ \mathcal{L}_{\text{GCN}}(\mathcal{M}(\boldsymbol{A}', \boldsymbol{X}; \boldsymbol{\Theta}), \boldsymbol{y}),$$

where $\mathcal{P}_\Delta^\mathcal{G}$ denotes the possible perturbations within budget $\Delta$. $\boldsymbol{y}$ is the ground-truth label. $\mathcal{L}_{\text{attack}}$ and $\mathcal{L}_{\text{GCN}}$ denote the loss of adversarial attack and model $\mathcal{M}$, respectively. $\mathcal{M}(\boldsymbol{A}', \boldsymbol{X}; \boldsymbol{\Theta}^*)$ refers to the prediction of model $\mathcal{M}$ on the poisoned graph $\mathcal{G}'$, where the weights $\boldsymbol{\Theta}^*$ keep trained to optimal.

We denote $\mathcal{T}$ as the set of target nodes aimed by the attacker. We consider the following two types of attacks: 1) **Targeted attacks.** The attacker aims to destroy prediction for a specific node $u$ by manipulating the adjacent edges of $u$ [70, 55, 9], and $\mathcal{T} = \{u\}$ in this situation. 2) **Non-targeted attacks.** The attacker aims to degrade the overall classification performance but does not care about which node is being targeted [71, 60]. Under this setting, $\mathcal{T} = \mathcal{V}_{\text{test}}$, where $\mathcal{V}_{\text{test}}$ is the full test set.

## 4 Methodology

### 4.1 Problem Formulation

In a poisoning attack, the training data from $\mathcal{G}$ is perturbed before fed into a GCN model $\mathcal{M}$, so that the performance of $\mathcal{M}$ can be degraded. Therefore, we formulate the defense problem as:

**Problem 1.** *Given a perturbed graph $\mathcal{G}'$ generated by a poisoning attacker, the aim of defense on target set $\mathcal{T}$ is to produce a more robust model $\mathcal{M}^r$ to optimize:*

$$\min_{\Theta^{r*}} \sum_{u \in \mathcal{T}} \|\mathcal{M}_u^r(\boldsymbol{A}', \boldsymbol{X}; \Theta^{r*}) - \mathcal{M}_u(\boldsymbol{A}, \boldsymbol{X}; \Theta^*)\|, \tag{2}$$

*where $\mathcal{M}_u^r(\boldsymbol{A}', \boldsymbol{X}; \Theta^{r*}) = \hat{\boldsymbol{y}}_u^r$ refers to the prediction of node $u$, when $\mathcal{M}^r$ is trained on the poisoned graph $\mathcal{G}'$ with optimal $\Theta^{r*}$. $\mathcal{M}_u(\boldsymbol{A}, \boldsymbol{X}; \Theta^*) = \hat{\boldsymbol{y}}_u$ is a hypothetical prediction that $\mathcal{M}$ would make if the clean graph $\mathcal{G}$ is accessible.*

In Eq. (2), we aim to force $\hat{\boldsymbol{y}}_u^r$ to approximate $\hat{\boldsymbol{y}}_u$, then the prediction $\hat{\boldsymbol{y}}_u$ in the benign situation can be recovered. However, the clean graph $\mathcal{G}$ is almost impossible to be accessed under the setting of adversarial attacks, which implies that we can not obtain $\mathcal{M}'$ by directly optimizing Eq.(2). To tackle this problem, we begin with investigating the potential impacts of structural attacks by examining the variability between $\boldsymbol{A}$ and $\boldsymbol{A}'$.

### 4.2 Spectral View of Structural Attacks

Following [14, 15], we observe the eigenvalues of $\hat{\boldsymbol{A}}$ after structural attacks as shown in Figure 1.

**Observation 1.** Under structural attacks, the high-valued eigenvalues (low-frequency components) of $\hat{\boldsymbol{A}}$ are likely to be more *robust* than the low-valued eigenvalues (high-frequency components).

With this empirical inspiration, GCN-SVD [14] directly preprocesses the poisoned graph and retains the top low-frequency components for defense. However, we argue that this justification is inaccurate. The relationship between robustness and graph spectrum is more complex than empirical observation. In the following, we provide an in-depth analysis of structural perturbations from the spectral domain.

#### 4.2.1 One-edge Perturbation

We first consider the targeted attacks with one-edge perturbation, where $\mathcal{T} = \{u\}$ and $\Delta = 1$. Note that under one-edge perturbation, *i.e.*, we conduct a single edge flip $e_{uv}$ on $\boldsymbol{A}$, $\Delta \boldsymbol{A}$ is a matrix with only 2 non-zero elements, namely $\Delta \boldsymbol{A}_{uv} = \Delta \boldsymbol{A}_{vu} = 1 - 2\boldsymbol{A}_{uv}$. Following [38], we assume that the eigenvector matrix $\boldsymbol{U}$ satisfies the following assumption:

**Assumption 1.** $\boldsymbol{U}$ has an orthonormal basis $(\boldsymbol{u}_y)_{y \in [N]}$ that consists of non-negative vectors, where $N$ is the number of nodes in the graph.

Then, we introduce the Lemma 1 to describe the differences of eigenvalues after structural attacks.

**Lemma 1.** *Given a graph $\mathcal{G} = (\mathcal{V}, \mathcal{E}, \boldsymbol{X})$ and one edge $e_{uv}$ to be perturbed, the change of $y$-th eigenvalue after perturbation is formulated as*

$$\Delta \lambda_y = \begin{cases} 2\boldsymbol{u}_{yu} \cdot \boldsymbol{u}_{yv} - \lambda_y(\boldsymbol{u}_{yu}^2 + \boldsymbol{u}_{yv}^2) & \text{if } \mathcal{E} \cup \{e_{uv}\} \\ -2\boldsymbol{u}_{yu} \cdot \boldsymbol{u}_{yv} + \lambda_y(\boldsymbol{u}_{yu}^2 + \boldsymbol{u}_{yv}^2) & \text{if } \mathcal{E} \setminus \{e_{uv}\}, \end{cases}$$

*where $\boldsymbol{u}_{yu}$ refers to the $u$-th element of eigenvector $\boldsymbol{u}_y$.*

The proof is available in the Appendix. With the help of Lemma 1, we can establish the relationship between robustness of GCNs and the change on each eigenvalue $|\Delta \lambda_y|$ resulting from perturbations. Intuitively, smaller changes on eigenvalues indicate stronger robustness of GCNs against attacks.

**Theorem 1 (Low-frequency).** *Assume that $\boldsymbol{u}_{au}^2 + \boldsymbol{u}_{av}^2 \neq 0$, $\boldsymbol{u}_{bu}^2 + \boldsymbol{u}_{bv}^2 \neq 0$, and*

$$(\boldsymbol{u}_{au} + \boldsymbol{u}_{av})^2 > (\boldsymbol{u}_{bu} - \boldsymbol{u}_{bv})^2. \tag{3}$$

*Then, there exists a pair of high-frequency $\lambda_a \in [-1, 0)$ and low-frequency $\lambda_b \in (0, 1]$, so that $|\Delta\lambda_a| > |\Delta\lambda_b|$ always holds if $\lambda_a$ and $\lambda_b$ satisfy*

$$\max\big(0, \frac{d_b - d_a + c_a\lambda_a}{c_b}\big) < \lambda_b < 1; \quad -1 < \lambda_a < \min\big(\frac{d_a - c_b + d_b}{c_a}, 0\big), \tag{4}$$

*where $c_a = \boldsymbol{u}_{au}^2 + \boldsymbol{u}_{av}^2$, $d_a = 2\boldsymbol{u}_{au} \cdot \boldsymbol{u}_{av}$, $c_b = \boldsymbol{u}_{bu}^2 + \boldsymbol{u}_{bv}^2$, and $d_b = 2\boldsymbol{u}_{bu} \cdot \boldsymbol{u}_{bv}$.*

The proof of Theorem 1 is available in the Appendix. From Theorem 1, we summarize two remarks.

**Remark 1.** The top low-frequency components (*i.e.*, components are selected from the values closer to 1) could be more robust than the top high-frequency ones (*i.e.*, components are selected from the values closer to -1) against one-edge perturbation under the condition from Inequality 3, which is consistent with Observation 1. However, it is still vulnerable to such attacks that perturb the edges satisfying $(\boldsymbol{u}_{au} + \boldsymbol{u}_{av})^2 \leq (\boldsymbol{u}_{bu} - \boldsymbol{u}_{bv})^2$. This is contrary to the justification from GCN-SVD [14] and implies that not all low-frequency components are robust.

**Remark 2.** We do not consider two special situations that $\boldsymbol{u}_{au}^2 = \boldsymbol{u}_{av}^2 = 0$ or $\boldsymbol{u}_{bu}^2 = \boldsymbol{u}_{bv}^2 = 0$, because they conclude that $\Delta\lambda_a = 0$ or $\Delta\lambda_b = 0$, respectively. This implies no perturbation happens on $\lambda_a$ or $\lambda_b$, which is rare in practice and out of discussion.

### 4.2.2 Robust Interval of Low-Frequency

As Remark 1 from Theorem 1, Inequality 3 still makes the low-frequency components of $\hat{\boldsymbol{A}}$ suffer from certain attacks. To further get rid of this condition, we derive a new bound for low-frequency $\lambda_b$ by removing the dependency on $\lambda_a$ in Inequality 4 here, which reveals the *robust interval*:

**Theorem 2 (Robust interval).** *Assume that $\boldsymbol{u}_{au}^2 + \boldsymbol{u}_{av}^2 \neq 0$ and $\boldsymbol{u}_{bu}^2 + \boldsymbol{u}_{bv}^2 \neq 0$. There exists low-frequency $\lambda_b$, so that $|\Delta\lambda_a| > |\Delta\lambda_b|$ always holds for any $-1 < \lambda_a < 0$ if $\lambda_b$ satisfies*

$$\max\big(0, \frac{d_b - d_a + c_a\lambda_a}{c_b}\big) < \lambda_b < \min\big(\frac{d_b + d_a - c_a\lambda_a}{c_b}, 1\big).$$

The proof is available in the Appendix. Theorem 2 indicates the interval of $\lambda_b$ that is able for defense, which we denote as the *robust interval* of $\lambda_b$. In other words, when the low-frequencies fall into the robust interval, they are always more robust than the high-frequencies under one-edge perturbation.

### 4.2.3 Non-Targeted Perturbation

In this part, we consider a more complex and practical setting, the multi-edge non-targeted perturbation, where $\mathcal{T} = \mathcal{V}_{\text{test}}$ and $\Delta > 1$. The perturbation $\mathcal{P}$ on each node $u \in \mathcal{T}$ can be treated as multi-edge targeted attack. Note that the multi-edge targeted perturbation is a special case when $\mathcal{T} = \{u\}$. Specifically, we discuss the type of attacks that either consecutively inserting or deleting edges on the clean graph, since every state after perturbations can be decomposed into consecutively inserting then removing edges. In this situation, the attacker directly manipulates the adjacent edges of $u$. Based on Theorem 1, we obtain the following results:

**Corollary 1 (Low-frequency).** *We decompose non-targeted attacks as two steps: 1) selecting target $u$ from $\mathcal{T}$, and 2) conducting perturbations on $u$ as multi-edge targeted attack. Assume $p$ adversarial edges are either consecutively inserted or deleted on node $u$ as set $\mathcal{P}_u$, i.e., $\mathcal{E} \cup \mathcal{P}_u$ or $\mathcal{E} \setminus \mathcal{P}_u$. With the same assumptions as in Theorem 1, there exists a pair of high-frequency $\lambda_a$ and low-frequency $\lambda_b$ so that $|\sum_{u \in \mathcal{T}} \sum_{v \in \mathcal{P}_u} \Delta\lambda_{auv}| > |\sum_{u \in \mathcal{T}} \sum_{v \in \mathcal{P}_u} \Delta\lambda_{buv}|$ always holds if $\lambda_a$ and $\lambda_b$ satisfy*

$$\max_{v \in \mathcal{P}_u, u \in \mathcal{T}}\big(0, \frac{d_{buv} - d_{auv} + c_{auv}\lambda_a}{c_{buv}}\big) < \lambda_b < 1; \ -1 < \lambda_a < \min_{v \in \mathcal{P}_u, u \in \mathcal{T}}\big(\frac{d_{auv} - c_{buv} + d_{buv}}{c_{auv}}, 0\big),$$

*where $c_{auv} = \boldsymbol{u}_{au}^2 + \boldsymbol{u}_{av}^2$, $d_{auv} = 2\boldsymbol{u}_{au} \cdot \boldsymbol{u}_{av}$, $c_{buv} = \boldsymbol{u}_{bu}^2 + \boldsymbol{u}_{bv}^2$, and $d_{buv} = 2\boldsymbol{u}_{bu} \cdot \boldsymbol{u}_{bv}$.*

Then we can have the robust interval under non-targeted perturbations with the help of Corollary 1 as:

**Corollary 2 (Robust interval).** *With same assumptions as in Theorem 2, $|\sum_{u \in \mathcal{T}} \sum_{v \in \mathcal{P}_u} \Delta\lambda_{auv}| > |\sum_{u \in \mathcal{T}} \sum_{v \in \mathcal{P}_u} \Delta\lambda_{buv}|$ always holds for any $-1 < \lambda_a < 0$ if $\lambda_b$ satisfies*

$$\max_{v \in \mathcal{P}_u, u \in \mathcal{T}}\big(0, \frac{d_{buv} - d_{auv} + c_{auv}\lambda_a}{c_{buv}}\big) < \lambda_b < \min_{v \in \mathcal{P}_u, u \in \mathcal{T}}\big(\frac{d_{buv} + d_{auv} - c_{auv}\lambda_a}{c_{buv}}, 1\big).$$

The proofs for Corollary 1 and Corollary 2 can be found in the Appendix.

**Remark 3.** Under non-targeted perturbation, one could come to the same remarks as Theorem 1 when consecutively inserting or deleting edges from the clean graph. The mixed situation will be left as a future work. Due to space constraints, please refer to Corollary 3 and Corollary 4 in the Appendix for details upon multi-edge targeted attack.

To summarize, we provide the theoretical justifications about the robustness of low-frequency components under the structural attacks in this section. These results imply that we can enhance the robustness of GCNs by utilizing the robust interval of low frequencies in $\hat{A}'$ of the poisoned graph.

### 4.3 GCN-LFR: General Robust Training Paradigm

Inspired by the theoretical findings in Section 4.2, it is natural to migrate the information from low-frequency components within robust interval to enhance the robustness of model $\mathcal{M}$. In this vein, we present GCN-LFR (**L**ow-**F**requency based **R**egularization), which introduces an auxiliary network $\mathcal{M}_{\text{LFR}}$ to extract the low-frequency information within robust interval to regularize $\Theta$ during the training procedure via parameter sharing [17]. Figure 2 depicts the whole framework of GCN-LFR.

#### 4.3.1 Auxiliary Regularization Net $\mathcal{M}_{\text{LFR}}$

The effectiveness of the parameter sharing based regularization is demonstrated by co-training an auxiliary neural network on various of Deep Neural Networks [2, 52]. In GCN-LFR, we propose our auxiliary regularization net $\mathcal{M}_{\text{LFR}}(A', X; \Theta)$ by explicitly designing the graph convolutional filter from Eq.(1) with the robust low-frequency components. However, explicitly computing the robustness interval is infeasible before the structural attack. To alleviate this chicken or the egg style causality dilemma, we employ a set of learnable parameters $F$ as filters to learn the robust interval for the low-frequency components during training.

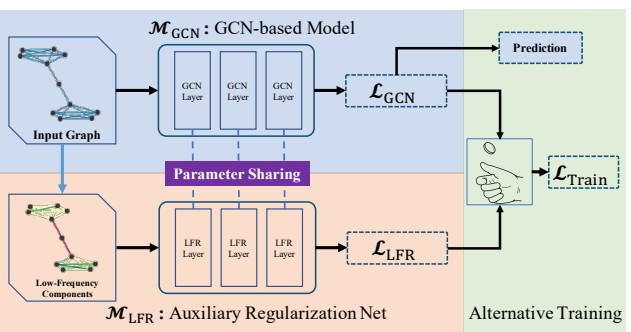

Figure 2: Overview of GCN-LFR. For any given GCN-based model $\mathcal{M}$, we enhance its robustness by co-training an auxiliary net $\mathcal{M}_{\text{LFR}}$ that utilizes only the low-frequency components as regularization.

Overall, the layer-wise update rule for $\mathcal{M}_{\text{LFR}}(A', X; \Theta, F)$ is:

$$\text{feature transformation} : H'^{(l)} = H^{(l)}\Theta, \quad \text{graph convolution} : H^{(l+1)} = \sigma(U'_{\text{low}}FU'^{\top}_{\text{low}}H'^{(l)}),$$

where $U'_{\text{low}}$ are the top-$k$ low-frequency eigenvectors from poisoned graph $\mathcal{G}'$, and the parameters $\Theta$ will be co-trained and shared to the original GCN model $\mathcal{M}_{\text{GCN}}(A', X; \Theta)$. $F = \text{diag}(f_1, \cdots, f_k)$ is a diagonal matrix with $k$ parameters as the graph filter, which can be learned in an end-to-end manner. Aside from the flexibility of learned $F$ falling into the robust interval, $F$ can also enlarge the capacity of $\mathcal{M}_{\text{LFR}}$ by making the learned embeddings $H$ generalize into the subspace span by $U'_{\text{low}}$.

#### 4.3.2 Co-Training Framework

Consider a learning task on graph $\mathcal{G}$, and a given GCN-based model $\mathcal{M}_{\text{GCN}}(A', X; \Theta)$ parametrized by $\Theta$ and poisoned by $A'$ is utilized to deal with this task. We denote the loss function of $\mathcal{M}_{\text{GCN}}$ as $\mathcal{L}_{\text{GCN}} = \mathcal{L}(\mathcal{M}_{\text{GCN}}(A', X; \Theta), y)$. Instead of minimizing $\mathcal{L}_{\text{GCN}}$ alone, we propose to co-train it with the same supervised loss on an auxiliary regularization net $\mathcal{L}_{\text{LFR}} = \mathcal{L}(\mathcal{M}_{\text{LFR}}(A', X; \Theta, F), y)$. In this way, we are able to compel the original model to learn a more reliable representation from the robust low-frequency components:

$$\mathcal{L}_{\text{total}} = (1 - \alpha)\mathcal{L}_{\text{GCN}} + \alpha\mathcal{L}_{\text{LFR}}, \tag{5}$$

where $\alpha \in [0, 1]$ is the weight coefficient for balancing the contribution of two losses. The transformation of the robust information from low-frequency components is achieved by regularizing the training of $\mathcal{L}_{\text{GCN}}$ with the shared parameters $\Theta$.

Instead of jointed optimizing $\mathcal{L}_{\text{GCN}}$ and $\mathcal{L}_{\text{LFR}}$, we exploit the idea of alternative training [52] to optimize the two losses. Specifically, at each epoch, we randomly choose one of the two losses in GCN-LFR to optimize with a probability determined by $\alpha$. To put it another way, we sample a random value $z \in U(0, 1)$, and $\alpha$ acts as a threshold that decides which loss would be optimized at each epoch. The thorough description of algorithm is depicted in the Appendix.

As compared to joint training, this alternative training scheme can reduce the computational cost and accelerate the training process. Furthermore, in practice, we observe that the generalization ability of model is also improved according to the alternative training strategy.

For the inference procedure, we only transfer the robustness of low-frequency components by cotraining $M_{LFR}$ and sharing its parameters during training procedure. Then we only use the prediction of the given model $M_{GCN}$ for inference. In other words, the extra complexity of our proposed method only from the training procedure, while the inference procedure stays the same, which further reflects the scalability of our proposed approach.

### 4.3.3 Complexity Analysis

We discuss the complexity of GCN-LFR here by the light example of VanillaGCN as backbone. For an input graph with $N$ nodes, $E$ edges, and $M$ as feature dimension on each node, the complexity of a solo VanillaGCN is $O(EM + NM^2)$ [57]. Then, with the help of Lanczos algorithm [28], the time complexity of calculating the top-$k$ eigen-pairs is $O(ckN)$, where $c$ is the average number of non-zero entries in a row of $\hat{A}$. Thus the overall complexity of GCN-LFR on VanillaGCN is $O(EM + N(M^2 + ck))$. In our experiments, we choose to tune the hyperparameter $k = d\%N$ as the proportion of low-frequency from grid search, in order to make GCN-LFR more adaptive to different graphs. In this case, the overall complexity would be $O(EM + NM^2 + cN^2))$. In a nutshell, the extra complexity introduced by GCN-LFR during co-training is not a burden, especially when the graph is sparse, as it is in most real-world applications.

## 5 Experiments

### 5.1 Experimental Setup

**Datasets.** We mainly focus on five node classification benchmark datasets under the semi-supervised setting: 1) three citation networks, Cora [37], Citeseer and Pubmed [43], 2) a coauthor network Coauthor CS [44], and 3) a co-purchase network Amazon Photo [36]. More details, including dataset description and statistical information, are deferred to the Appendix.

**Adversarial samples generation.** We evaluate our model under three kinds of adversarial attack settings from the existing studies: *targeted attack with one-edge perturbation* (Nettack-One [70]), *targeted attack with multi-edge perturbation* (Nettack-Multi [70]), and *non-targeted attack* (Mettack [71]). Details upon the three adversarial attack settings can be found in the Appendix.

**Target GCN-based models.** We apply GCN-LFR to five GCN-basd models to extensively evaluate its defense performance under three attack settings. The five models are: VanillaGCN [27], JK-Net [61], SGC [54], Graph-U-Net [16], and GCNII [10].

**Defense baselines.** We compare GCN-LFR with four SOTA general graph defenders with their available public implementations. GCN-Jaccard [55] and GCN-SVD [14] are two preprocessing based defending methods. GNNGUARD [63] and Pro-GNN [25] are two spatial-based approaches. Details upon hyperparameters and architectures can be referred to in the Appendix.

### 5.2 Defense Results Against Targeted and Non-Targeted Attacks

**Targeted attack with one-edge perturbation.** We first examine the robustness of GCN-LFR against targeted one-edge perturbation under Nettack-One and report the results in Table 1. The column "Attacked" shows the performance without any defense. From this column, we can find that Nettack-One can successfully damage the performance of various models even with only a single edge flip. However, GCN-LFR can dramatically defend given GCNs against Nettack-One and consistently outperforms state-of-the-art baselines. Remarkably, we observe that the low-rank preprocessing

Table 1: Defense performance (multi-class classification accuracy) against one-edge targeted attacks under Nettack-One. **Note:** the accuracy is 1 on clean graph since only the correctly classified nodes are targeted.

| Model | Dataset | Attacked | GCN-Jaccard | GCN-SVD | GNNGUARD | GCN-LFR |
|---|---|---|---|---|---|---|
| VanillaGCN [27] | Cora | 0.350 | 0.675 | 0.700 | 0.775 | **0.800** |
| | Citeseer | 0.425 | 0.625 | 0.825 | 0.725 | **0.775** |
| | Pubmed | 0.450 | 0.550 | 0.600 | 0.775 | **0.825** |
| | Coauthor CS | 0.650 | 0.675 | 0.725 | 0.825 | **0.875** |
| | Amazon Photo | 0.500 | 0.525 | 0.625 | 0.650 | **0.750** |
| JK-Net [61] | Cora | 0.350 | 0.575 | 0.675 | 0.775 | **0.825** |
| | Citeseer | 0.450 | 0.650 | 0.550 | 0.750 | **0.800** |
| | Pubmed | 0.350 | 0.375 | 0.625 | 0.800 | **0.850** |
| | Coauthor CS | 0.600 | 0.625 | 0.675 | 0.725 | **0.775** |
| | Amazon Photo | 0.550 | 0.575 | 0.675 | 0.675 | **0.800** |
| SGC [54] | Cora | 0.425 | 0.675 | 0.550 | 0.750 | **0.825** |
| | Citeseer | 0.475 | 0.600 | 0.650 | 0.725 | **0.750** |
| | Pubmed | 0.300 | 0.475 | 0.650 | 0.750 | **0.800** |
| | Coauthor CS | 0.600 | 0.650 | 0.725 | 0.750 | **0.775** |
| | Amazon Photo | 0.550 | 0.575 | 0.600 | 0.675 | **0.725** |
| Graph-U-Net [16] | Cora | 0.375 | 0.525 | 0.650 | 0.725 | **0.800** |
| | Citeseer | 0.275 | 0.350 | 0.575 | 0.750 | **0.775** |
| | Pubmed | 0.475 | 0.550 | 0.675 | 0.700 | **0.750** |
| | Coauthor CS | 0.625 | 0.600 | 0.700 | 0.775 | **0.850** |
| | Amazon Photo | 0.500 | 0.525 | 0.650 | 0.725 | **0.825** |
| GCNII [10] | Cora | 0.475 | 0.650 | 0.575 | 0.825 | **0.875** |
| | Citeseer | 0.550 | 0.600 | 0.625 | 0.725 | **0.750** |
| | Pubmed | 0.400 | 0.475 | 0.675 | 0.750 | **0.775** |
| | Coauthor CS | 0.625 | 0.650 | 0.700 | 0.800 | **0.825** |
| | Amazon Photo | 0.650 | 0.425 | 0.600 | 0.675 | **0.775** |

approach GCN-SVD is not very effective, which implies that not all low-frequency components are robust, and our strategy by transferring robust knowledge with learnable $F$ is beneficial.

Table 2: Defense performance (multi-class classification accuracy) against multi-edge targeted attacks under Nettack-Multi. **Note:** the accuracy is 1 on clean graph since only the correctly classified nodes are targeted.

| Model | Dataset | Attacked | GCN-Jaccard | GCN-SVD | GNNGUARD | GCN-LFR |
|---|---|---|---|---|---|---|
| GCNII [10] | Cora | 0.160 | 0.540 | 0.585 | 0.695 | **0.700** |
| | Citeseer | 0.170 | 0.370 | 0.590 | **0.710** | 0.655 |
| | Pubmed | 0.080 | 0.275 | 0.430 | 0.605 | **0.650** |
| | Coauthor CS | 0.060 | 0.350 | 0.525 | 0.730 | **0.775** |
| | Amazon Photo | 0.120 | 0.440 | 0.620 | 0.715 | **0.720** |

Table 3: Defense performance (multi-class classification accuracy) against non-targeted attacks under Mettack.

| Model | Dataset | Attacked | GCN-Jaccard | GCN-SVD | GNNGUARD | Pro-GNN | GCN-LFR |
|---|---|---|---|---|---|---|---|
| VanillaGCN [27] | Cora | 0.701 | 0.760 | 0.717 | 0.764 | 0.773 | **0.777** |
| | Citeseer | 0.657 | 0.680 | 0.664 | 0.687 | **0.693** | 0.690 |
| | Pubmed | 0.807 | 0.845 | 0.823 | 0.848 | 0.852 | **0.855** |
| | Coauthor CS | 0.836 | 0.865 | 0.852 | 0.868 | - | **0.875** |
| | Amazon Photo | 0.854 | 0.877 | 0.862 | 0.882 | - | **0.887** |

**Targeted attack with multi-edge perturbation.** Next, we evaluate the defending performance of targeted attacks with multi-edge perturbation. We report the defending performance on the newest variant, GCNII, in Table 2. It is not surprising that GCNII suffers more serious performance degradation under Nettack-Multi than that of Nettack-One. However, GCN-LFR remains the best defender among all baselines even when the amount of perturbed edges increases. We also observe that GCN-LFR achieves better defense under one-edge perturbation than multi-edge. We suppose this is due to the wider robust interval in Theorem 2, since the bounds are tighter when iterating over all candidate edges in Corollary 4.

**Non-targeted attack.** To validate the defending performance of GCN-LFR against non-targeted attacks, we utilize Mettack to conduct adversarial edges globally. We also include the comparison

with an additional baseline Pro-GNN under this setting on small datasets since Pro-GNN is extremely time-consuming on large datasets. Table 3 demonstrates the results on VanillaGCN for all datasets. In Table 3, we can observe that the defending performance of GCN-LFR is also comparable with all baselines.

Table 4: Classification accuracy on benign (*i.e.*, non-attacked) datasets with and without GCN-LFR.

| Bengin dataset | Cora | | Citeseer | | Pubmed | | Coauthor CS | | Amazon Photo | |
|---|---|---|---|---|---|---|---|---|---|---|
| w or w/o LFR | w/o | w | w/o | w | w/o | w | w/o | w | w/o | w |
| VanillaGCN | 0.815 | 0.818 | 0.721 | 0.701 | 0.863 | 0.858 | 0.907 | 0.911 | 0.921 | 0.925 |
| JK-Net | 0.820 | 0.827 | 0.705 | 0.693 | 0.854 | 0.836 | 0.927 | 0.891 | 0.929 | 0.896 |
| SGC | 0.825 | 0.819 | 0.719 | 0.701 | 0.859 | 0.861 | 0.928 | 0.897 | 0.929 | 0.927 |
| Graph-U-Net | 0.825 | 0.838 | 0.697 | 0.688 | 0.835 | 0.842 | 0.926 | 0.907 | 0.925 | 0.918 |
| GCNII | 0.812 | 0.813 | 0.695 | 0.698 | 0.864 | 0.838 | 0.931 | 0.904 | 0.934 | 0.927 |

## 5.3 Ablation Studies

**Performance of** GCN-LFR **on clean datasets.** Aside from the defense against adversarial attacks, a successful defender needs not to affect the performance on clean datasets. Table 4 shows the performance comparison of five GCN models with and without regularization net $\mathcal{M}_{\text{LFR}}$ on five benign (*i.e.*, non-attacked) datasets. As shown in Table 4, GCN-LFR achieves a quite competitive classification accuracy and even better in some cases. This indicates that GCN-LFR does not harm the performance of GCNs when no attack happens while successfully dealing with the poisoned graphs. We also conduct T-test on the classification accuracy over all benign datasets w or w/o LFR. which can be found in the Appendix.

**In-depth analysis on** $\mathcal{M}_{\textbf{LFR}}$ **net.** To justify the necessity of each component of GCN-LFR, we implement several variants of GCN-LFR with VanillaGCN and evaluate the performance on the benign Cora dataset. Figure 3 shows the comparison among GCN-LFR and other variants on the training and validation accuracy. GCN and LFR represent training with only VanillaGCN and $\mathcal{M}_{\text{LFR}}$ net, respectively. GCN-LFR$_{\text{Fix}}$ is a variant of GCN-LFR that replaces the learnable filter $\boldsymbol{F}$ with the truncated eigenvalues $\boldsymbol{\Lambda}'_{\text{low}}$. We have the following observations: 1) LFR net achieves lower training accuracy and suffers from severe underfitting issue. It indicates that training with only low-frequency components would lose information despite their robustness. 2) GCN-LFR achieves competitive performance against GCN. It re-confirms the results in Section 5.3 that GCN-LFR does not harm performance in the absence of attack. 3) GCN-LFR achieves substantially better validation accuracy than that of GCN-LFR$_{\text{Fix}}$. It implies that the trainable graph filter $\boldsymbol{F}$ is more flexible and can aid the generalization of learned embeddings in the subspace spanned by $\boldsymbol{U}'_{\text{low}}$.

Aside from Figure 3, we further provide the comparison among VanillaGCN, GCN-LFR$_{\text{Fix}}$ and GCN-LFR under both benign and non-targeted attack settings here for better and sufficient demonstration in Table 5. It shows that GCN-LFR has both better performance and robustness over VanillaGCN and GCN-LFR$_{\text{Fix}}$. Interestingly, the better generalization ability of GCN-LFR over GCN-LFR$_{\text{Fix}}$ could be observed implicitly either in this way.

Table 5: Classification accuracy comparison among VanillaGCN, GCN-LFR$_{\text{Fix}}$ and GCN-LFR on both benign and non-targeted attacked datasets.

| Dataset | Cora | | Citeseer | | Pubmed | | Coauthor CS | | Amazon Photo | |
|---|---|---|---|---|---|---|---|---|---|---|
| Methods | Benign | Attacked | Benign | Attacked | Benign | Attacked | Benign | Attacked | Benign | Attacked |
| VanillaGCN | 0.830 | 0.771 | 0.720 | 0.654 | 0.860 | 0.794 | 0.902 | 0.816 | 0.929 | 0.824 |
| GCN-LFR$_{\text{Fix}}$ | 0.779 | 0.726 | 0.683 | 0.634 | 0.832 | 0.786 | 0.883 | 0.808 | 0.876 | 0.793 |
| GCN-LFR | 0.824 | 0.791 | 0.706 | 0.670 | 0.856 | 0.802 | 0.912 | 0.837 | 0.928 | 0.943 |

**Joint optimization vs. alternate optimization on** GCN-LFR**.** We investigate the benefits of alternate optimization of GCN-LFR (*i.e.*, GCN-LFR$_{\text{Alter}}$). We denote the joint optimization of GCN-LFR as GCN-LFR$_{\text{Joint}}$. Figure 4 provides the results of comparison. The left-side of Figure 4 demonstrates the training and validation accuracy on Pubmed dataset. VanillaGCN is used as backbone here. We can observe that GCN-LFR$_{\text{Alter}}$ achieves better training and validation performance compared with GCN-LFR$_{\text{Joint}}$. Furthermore, the right-side of Figure 4 shows the acceleration ratio of GCN-LFR$_{\text{Alter}}$ over GCN-LFR$_{\text{Joint}}$ in terms of training time on five datasets. Acceleration ratio denotes the quotient obtained by dividing the time-consumed of GCN-LFR$_{\text{Joint}}$ by that of GCN-LFR$_{\text{Alter}}$. We can find

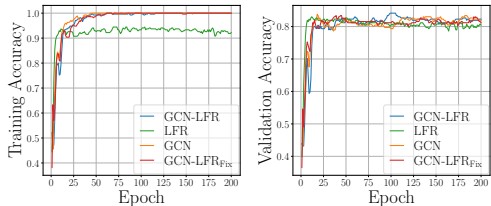
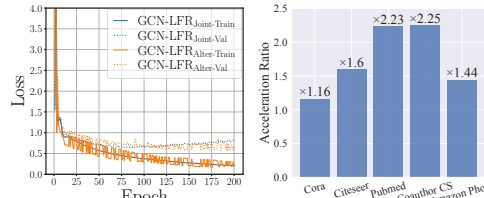

Figure 3: The training and validation accuracy comparison among GCN-LFR, two components of GCN-LFR, *i.e.*, VanillaGCN and LFR, and a variant of GCN-LFR, namely GCN-LFR$_{Fix}$.

Figure 4: **Left:** the training and validation loss comparison between GCN-LFR$_{Alter}$ on Pubmed with backbone VanillaGCN. **Right:** the acceleration ratio of GCN-LFR$_{Alter}$ over GCN-LFR$_{Joint}$.

that GCN-LFR$_{Alter}$ is more efficient by achieving an average of $\times 1.74$ times acceleration, since GCN-LFR$_{Alter}$ only needs to optimize one model while GCN-LFR$_{Joint}$ needs to handle two models at each epoch. It verifies that GCN-LFR$_{Alter}$ enjoys both effectiveness and efficiency.

Aside from Figure 4, we also provide the train/validation/test accuracy comparison between GCN-LFR$_{Joint}$ and GCN-LFR$_{Alter}$ over all datasets in Table 6 for clearer demonstration. The consistent conclusion as above can be drawn from Table 6.

Table 6: Train/validation/test accuracy comparison between GCN-LFR$_{Joint}$ and GCN-LFR$_{Alter}$.

| Dataset | Cora | Citeseer | Pubmed | Coauthor CS | Amazon Photo |
|---|---|---|---|---|---|
| Joint-Train | **0.999 ± 0.001** | 0.991 ± 0.004 | 0.929 ± 0.002 | **0.986 ± 0.004** | 0.975 ± 0.007 |
| Alter-Train | **0.999 ± 0.001** | **0.995 ± 0.002** | **0.932 ± 0.013** | 0.977 ± 0.006 | **0.986 ± 0.005** |
| Joint-Val | 0.827 ± 0.018 | 0.702 ± 0.027 | 0.832 ± 0.006 | 0.903 ± 0.008 | 0.939 ± 0.004 |
| Alter-Val | **0.837 ± 0.025** | **0.717 ± 0.008** | **0.843 ± 0.008** | **0.907 ± 0.007** | **0.942 ± 0.009** |
| Joint-Test | 0.817 ± 0.011 | 0.698 ± 0.007 | 0.846 ± 0.005 | 0.908 ± 0.003 | 0.917 ± 0.001 |
| Alter-Test | **0.833 ± 0.009** | **0.701 ± 0.011** | **0.853 ± 0.010** | **0.912 ± 0.008** | **0.931 ± 0.015** |

**Eigenvalues vs. learned $F$ in the robust interval.** To evaluate whether the proposed theoretical analysis is reliable, we check how many $\lambda_b$ and the trained $F$ fall into the robust interval on all datasets here. We report the results of top 25% smallest of eigenvalues as $\lambda_b$ on the same 40 target nodes after one-edge perturbation in Figure 5. We can find that, though by choosing a considerably larger proportion as low-frequency, we still have an average of around 40% of $\lambda_b$ falling into the robust interval. Further, the learnable $F$ has more values falling into the robust interval after training, which is consistent with our observation in Figure 3. It also verifies that $F$ could potentially learn the frequencies within the robust interval for low-frequency components and lead to a more robust model.

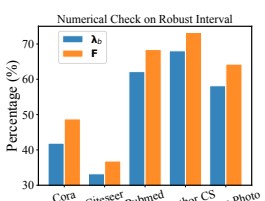

Figure 5: Numerical check on the robust interval for all datasets.

## 6 Conclusion and Limitations

In pursuit of developing defending methods for the adversarial attacks on GCNs, we present the first vulnerability study between the graph spectrum and the robustness behaviour of GCNs, which indicates that the low-frequency components in graphs could be more robust than the high-frequency ones under certain conditions. Moreover, we further verify a robust interval of the low-frequency components, which implies that not all low-frequency components are robust unless they fall within the robust interval. Guided by this theory, we propose a general robust training paradigm, dubbed GCN-LFR, that suggests training an auxiliary network jointly with the original model through parameter sharing to transfer the robustness of low-frequency components. Extensive experimental verifications demonstrate the resistance of GCN-LFR when being exposed to adversarial attacks.

Currently, we only consider the purification in the context of structural perturbations. Given that adversarial perturbations will occur at the level of features and nodes, we would like to expand the proposed architecture in the future to protect against various forms of adversarial attacks. Meanwhile, we make every effort to include a thorough experimental study. Further results on possible datasets and baselines will be added as a future work, including the extension to large scale datasets such as OGB [21]. Societal impacts are discussed in the Appendix.

## Acknowledgments and Disclosure of Funding

Funding in direct support of this work: National Key Research and Development Program of China No. 2020AAA0107801, and National Natural Science Foundation of China (No. 62050110, No. 62102222).

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
