# Appendix: Not All Low-Pass Filters are Robust in Graph Convolutional Networks

## Contents


# B    Broader Impact

Graph Convolutional Networks (GCNs) could be crucial tools for a broad range of applications, including social networks, computer vision, natural language processing, traffic prediction, chemistry, protein design, recommendation system and so on [64, 58].

Any of these applications may have a different social effect. The use of GCNs could improve protein design efficiency and lead to the development of new medicines, but it could also result in job losses. Though traffic prediction would undoubtedly assist us in better planning our journeys, the processing of personal data can pose a challenge. Another major problem is that while these graph responsive assessment models can have the ability to improve results, security concerns often hinder their practical implementation.

Since the symmetric normalized adjacency matrix is widely chosen as the fundamental building block of current architectures, the GCN family will profit greatly from the key outcome of our work, given the societal consequences. Meanwhile, our suggested co-training model GCN-LFR protects against adversarial threats while maintaining precision in benign situations. With the support of GCN-LFR, the use of GCNs in safety-critical applications (such as power grids, financial risk control, and drug screening) can be extended.

# C    Additional Related Work

Adversarial attacks have become an intriguing direction relating to the behavior of neural networks [62, 59]. The attackers can insert slight perturbation on samples, which is usually unnoticeable to a human, and the output of neural networks will be completely fooled. While most current works focus on dealing with the grid or sequential data, adversarial attack and defense on graph neural networks have

drawn unprecedented attention from researchers recently [47, 24, 49]. **Defenses on graphs.** While GCNs have already shown promising results on various graph-related tasks, fewer efforts are put into the enhancement of robustness of GCNs, compared with the rapid progress dealing with images or texts. We briefly introduce the SOTA defense efforts [67, 55, 5, 69, 63, 56, 23] for GCNs here.

- **GCN-Jaccard** [55] proposes to examine the fake edges in the step of preprocessing of GCNs by utilizing similarity metrics. [67, 25, 23] focus on proposing new variants of GCN that can effectively defend against adversarial attack, but they are limited to VanillaGCN and challenging to be employed upon other GCNs. [5, 69] focus on theoretically certifying the robustness of GCNs under perturbations of the graph structure and node features instead of defense mechanisms. Meanwhile, [56] provides the first information-theoretic principle inherited from the information bottleneck principle for supervised representation learning on graph-structured data. However, we aim to provide a practical adversarial defense framework with theoretical explanations.
- **GNNGUARD** [63] detects the adverse effects existing in the relationship between the graph structure and node features by neighbor importance estimation and layer-wise graph memory. Though GNNGUARD [63] can be incorporated with many GCN models, the approach is derived from the spatial network theory of homophily. In contrast, our GCN-LFR method considers this problem from the perspective of spectral domain. A similar approach to GNN-GUARD is Low-Pass [53], which also aims to eliminate the side effects from neighborhoods by considering low-pass message passing. Given neither public implementation nor source code upon request is available, we choose not to include Low-Pass in our baselines.
- **GCN-SVD** [14] finds that only the high-rank singular components of the graph are affected by the attacking method Nettack [70]. Then GCN-SVD [14] suggests that the power of Nettack can be greatly reduced if a low-rank approximation of the graph is utilized in contrast to the original clean graph.

The most related work to this paper is GCN-SVD. However, they only provide empirical observations on the robustness of low-rank approximation, and use SVD decomposition to purify the poisoned graph with only the structural information. In this paper, we first build the theoretical connection between robustness and graph spectrum, then propose a framework integrating both graph structure and node features to enhance robustness.

**Adaptive attacks.** In the computer vision community, there has recently been a new branch for adaptive attacks on defending models [51]. While adaptive adversarial examples can exist in graphs, there are currently no such works in the graph learning sector. The defense against adaptive attacks on graph structured data is also interesting and significant, but it is beyond the reach of this research and will be left as a future work.

# D    Additional Preliminaries on Graph Signal Filtering

**Graph frequency from GSP.** Graph Signal Processing (GSP) extends the concept of frequency in Discrete Signal Processing and focuses on the analysis and processing of data points whose relations are modeled as a graph [45, 39]. Let $\mathcal{L} = U\Lambda U^\top$ be the eigen-decomposition of the symmetric normalized Laplacian $\mathcal{L}$, where $U = [u_1^\top, \cdots, u_N^\top] \in \mathbb{R}^{N \times N}$ and $\Lambda = \text{diag}(\lambda_1, \cdots, \lambda_N)$ are the eigenvectors and eigenvalues of $\mathcal{L}$, respectively. The Laplacian matrix has a complete set of orthonormal eigenvectors $U$. The eigenvalues are sorted non-decreasingly as $0 \leq \lambda_1 \leq \lambda_2 \leq \cdots \leq \lambda_N < 2$. Therefore, one can treat each eigenvector $u$ as an oscillation pattern and its corresponding eigenvalue $\lambda$ as the frequency of the oscillation. In other words, the frequency of an eigenvector $u_i$ of the Laplacian matrix is its corresponding eigenvalue $\lambda_i$ [20]:

$$u_i^\top \mathcal{L} u_i = u_i^\top \lambda_i u_i = \lambda_i. \tag{6}$$

To be consistent with the Laplacian matrix-based definition of graph frequency, throughout this paper, we call the frequency components of Laplacian with *small eigenvalues* as low-frequency components and vice versa. Due to the connection between the symmetric normalized adjacency matrix $\hat{A} = I_n - \mathcal{L}$ and $\mathcal{L}$, the eigenvalues of $\hat{A}$ read $\Lambda_{\hat{A}} = I_n - \Lambda_{\mathcal{L}}$, which range from $-1$ to $1$. Specifically, the *low-frequency components* refer to frequency components of $\mathcal{L}$ with eigenvalues in $[0, 1)$ (of $\hat{A}$ with eigenvalues in $(0, 1]$), and the *high-frequency components* refer to frequency components of $\mathcal{L}$ with eigenvalues in $(1, 2]$ (of $\hat{A}$ with eigenvalues in $(-1, 0)$).

## E   Measure of Significant Change on the Spectrum

There are two ways to define the significant change in eigenvalues. One is the *absolute change* as we demonstrated in Figure 1, and the other one is the *rate of change* over eigenvalues before and after perturbation. We discuss the two ways here with respect to the influence on the reconstructed adjacency matrix $A'$. Considering the perturbation on the all frequencies as a diagonal matrix $\epsilon$, where $\epsilon_{ii}$ is the absolute change on $i_{th}$ eigenvalue. Thus for the original adjacency matrix $A = U\Lambda U^T$, the reconstructed adjacency matrix after perturbation by $\epsilon$ is approximated as $A' = U(\Lambda + \epsilon)U^T = U\Lambda U^T + U\epsilon U^T$. Thus the absolute change can better reflect the perturbation on the reconstructed $A'$, which is directly related to the analysis of robustness.

## F   The Results of Multi-edge Perturbation on Targeted Attacks

In this part, we give the results under the multi-edge targeted perturbation, where $\mathcal{T} = \{u\}$ and $\Delta > 1$. Specifically, we discuss the type of attacks that either consecutively inserting or deleting edges on the clean graph. In this situation, the attacker directly manipulates the adjacent edges of $u$.

**Corollary 3** (**Low-frequency**). *Assume that $p$ adversarial edges are either consecutively inserted or deleted on node $u$ from the clean graph, i.e., $\mathcal{E} \cup \mathcal{P}$ or $\mathcal{E} \setminus \mathcal{P}$, where $\mathcal{P}$ refers to the set of $p$ adversarial edges. Then, with the same assumptions as in Theorem 1, there exists a pair of $\lambda_a$ and $\lambda_b$, so that $|\sum_{v\in\mathcal{P}}\Delta\lambda_{av}| > |\sum_{v\in\mathcal{P}}\Delta\lambda_{bv}|$ always holds if $\lambda_a$ and $\lambda_b$ satisfy*

$$\max_{v\in\mathcal{P}}\left(0, \frac{d_{bv} - d_{av} + c_{av}\lambda_a}{c_{bv}}\right) < \lambda_b < 1; \quad -1 < \lambda_a < \min_{v\in\mathcal{P}}\left(\frac{d_{av} - c_{bv} + d_{bv}}{c_{av}}, 0\right),$$

*where $c_{av} = \boldsymbol{u}_{au}^2 + \boldsymbol{u}_{av}^2$, $d_{av} = 2\boldsymbol{u}_{au} \cdot \boldsymbol{u}_{av}$, $c_{bv} = \boldsymbol{u}_{bu}^2 + \boldsymbol{u}_{bv}^2$, and $d_{bv} = 2\boldsymbol{u}_{bu} \cdot \boldsymbol{u}_{bv}$.*

Then we have the robust interval under multi-edge perturbation as:

**Corollary 4** (**Robust interval**). *With the same assumptions from Theorem 2 and Corollary 3, $|\sum_{v\in\mathcal{P}}\Delta\lambda_{av}| > |\sum_{v\in\mathcal{P}}\Delta\lambda_{bv}|$ always holds for any $-1 < \lambda_a < 0$ if $\lambda_b$ satisfies*

$$\max_{v\in\mathcal{P}}\left(0, \frac{d_{bv} - d_{av} + c_{av}\lambda_a}{c_{bv}}\right) < \lambda_b < \min_{v\in\mathcal{P}}\left(\frac{d_{bv} + d_{av} - c_{av}\lambda_a}{c_{bv}}, 1\right). \tag{7}$$

The proofs of Corollary 3 and Corollary 4 can be found in Section H.4.

**Remark 4.** From Corollary 3 and Corollary 4 , we can come to the same remarks with one-edge perturbation that the low-frequency components of $\hat{A}$ from the robust interval could be more robust than the high-frequency ones when we successively insert or delete edges in the graph.

## G   Details of the Proposed Algorithm

In Algorithm 1, we take node classification as the example task. It is worth noting that GCN-LFR is a general robust training framework for defending any GCN-based model on various graph learning tasks such as graph classification and link prediction.

## H   Proofs and Derivations

### H.1   Proof of Lemma 1

We include two immediate results to prove Lemma 1. The perturbation on the eigenvalue of an undirected graph resulting from edge modification has been well studied in literature [66, 4, 9]. From [46], we have:

**Lemma 2.** *[4] $\lambda$ is an eigenvalue of $\boldsymbol{D}^{-1/2}\boldsymbol{A}\boldsymbol{D}^{-1/2} := \hat{A}$ with eigenvector $\hat{\boldsymbol{u}} = \boldsymbol{D}^{1/2}\boldsymbol{u}$ if and only if $\lambda$ and $\boldsymbol{u}$ solve the generalized eigen-problem $\boldsymbol{A}\boldsymbol{u} = \lambda\boldsymbol{D}\boldsymbol{u}$.*

According to Lemma 2, instead of solving $\hat{A}$ directly, we solve the generalized eigen-problem $\boldsymbol{A}\boldsymbol{u} = \lambda\boldsymbol{D}\boldsymbol{u}$ to obtain the eigenvalue of $\hat{A}$. Therefore, the eigenvalues of $\hat{A}$ after perturbation can be well approximated by the clean adjacency matrix $\boldsymbol{A}$ and corresponding eigen-pair before attack. For the approximation of the perturbed eigenvalue, we have:

**Algorithm 1** A training procedure of GCN-LFR for node classification tasks.

---

1: **Input:** A poisoned graph $\mathcal{G}'$ with adjacency matrix $\boldsymbol{A}'$ and feature matrix $\boldsymbol{X}$; The node labels $\boldsymbol{y}$; The GCN-based model $\mathcal{M}_{\text{GCN}}$; The regularization net $\mathcal{M}_{\text{LFR}}$; The number of epochs $T$; The weight coefficient $\alpha$.
2: **Output:** The model parameter $\boldsymbol{\Theta}$ of $\mathcal{M}_{\text{GCN}}$.
3: Initialize the models $\mathcal{M}_{\text{GCN}}$ and $\mathcal{M}_{\text{LFR}}$.
4: **for** $t = 1$ to $T$ **do**
5:     $z \sim \text{uniform}(0, 1)$ {*Generate a random number in* $[0, 1)$.}
6:     **if** $z \geq \alpha$ **then**
7:       $\mathcal{L} \leftarrow \mathcal{L}_{\text{GCN}}(\mathcal{M}_{\text{GCN}}(\boldsymbol{A}', \boldsymbol{X}; \boldsymbol{\Theta}), \boldsymbol{y})$
8:     **else**
9:       $\mathcal{L} \leftarrow \mathcal{L}_{\text{LFR}}(\mathcal{M}_{\text{LFR}}(\boldsymbol{A}', \boldsymbol{X}; \boldsymbol{\Theta}, \boldsymbol{F}), \boldsymbol{y})$
10:    **end if**
11:    Optimize $\mathcal{L}$.
12: **end for**
13: Return $\boldsymbol{\Theta}$ .

---

**Lemma 3.** *[46] We denote $\boldsymbol{A}' = \boldsymbol{A} + \Delta\boldsymbol{A}$ as a perturbed version of $\boldsymbol{A}$ by modifying edges, and $\Delta\boldsymbol{D}$ as the respective change in the degree matrix. $\hat{\boldsymbol{A}} = \boldsymbol{U}\boldsymbol{\Lambda}\boldsymbol{U}^{\top}$ is the eigen-decomposition of the symmetric normalized adjacency matrix $\hat{\boldsymbol{A}}$. $\lambda_y$ and $\boldsymbol{u}_y$ are the $y_{th}$ eigen-pair of eigenvalue and eigenvector of $\hat{\boldsymbol{A}}$ and also solve the generalized eigen-problem $\boldsymbol{A}\boldsymbol{u}_y = \lambda_y \boldsymbol{D}\boldsymbol{u}_y$. Then the perturbed eigenvalue $\lambda'_y$ can be calculated as:*

$$\lambda'_y = \lambda_y + (\boldsymbol{u}_y^{\top}\Delta\boldsymbol{A}\boldsymbol{u}_y - \lambda_y \boldsymbol{u}_y^{\top}\Delta\boldsymbol{D}\boldsymbol{u}_y) + \mathcal{O}(\|\Delta\boldsymbol{A}\|). \tag{8}$$

With Lemma 2 and Lemma 3, we now can conduct proof of Lemma 1:

*Proof.* **Lemma 1.** Let $\boldsymbol{e}_u$ be the vector of all zeros and a single one at position $u$. Then, we derive

$$\Delta\boldsymbol{A} = \Delta w_{uv}(\boldsymbol{e}_u\boldsymbol{e}_v^{\top} + \boldsymbol{e}_v\boldsymbol{e}_u^{\top}), \tag{9}$$

$$\Delta\boldsymbol{D} = \Delta w_{uv}(\boldsymbol{e}_u\boldsymbol{e}_u^{\top} + \boldsymbol{e}_v\boldsymbol{e}_v^{\top}), \tag{10}$$

where $\Delta w_{uv} = (1 - 2\boldsymbol{A}_{uv})$ indicates the edge flip, i.e $\pm 1$. Thus, by substituting Eq.(9) and Eq.(10) into Eq.(8), we obtain

$$\Delta\lambda_y = \pm \left(2\boldsymbol{u}_{yu} \cdot \boldsymbol{u}_{yv} - \lambda_y(\boldsymbol{u}_{yu}^2 + \boldsymbol{u}_{yv}^2)\right). \tag{11}$$

$\square$

### H.2    Proof of Theorem 1

*Proof.* **Theorem 1.** Since the perturbation contains two cases: edge insertion and edge deletion, we will discuss each case separately. We first denote $c_a = \boldsymbol{u}_{au}^2 + \boldsymbol{u}_{av}^2$, $d_a = 2\boldsymbol{u}_{au} \cdot \boldsymbol{u}_{av}$, $c_b = \boldsymbol{u}_{bu}^2 + \boldsymbol{u}_{bv}^2$, and $d_b = 2\boldsymbol{u}_{bu} \cdot \boldsymbol{u}_{bv}$.

**Case 1** (Insertion). One-edge is inserted to the graph, *i.e.*, $\mathcal{E} \cup e_{uv}$. Therefore,

$$\Delta\lambda_y = 2\boldsymbol{u}_{yu} \cdot \boldsymbol{u}_{yv} - \lambda_y(\boldsymbol{u}_{yu}^2 + \boldsymbol{u}_{yv}^2), \tag{12}$$

which implies that $\Delta\lambda_a = d_a - c_a\lambda_a$ and $\Delta\lambda_b = d_b - c_b\lambda_b$. Since we assume $\boldsymbol{u} \geq 0$ for all $\boldsymbol{u}$ from Assumption 1, thus for $-1 < \lambda_a < 0$, we can have $\Delta\lambda_a > 0$ for all $\lambda_a$, *i.e.*, $|\lambda_a| > 0$. Then we conduct category discussion on $\Delta\lambda_b$ with $0 < \lambda_b < 1$.

1. For $\Delta\lambda_b \geq 0$, we have $\lambda_b \leq \frac{2\boldsymbol{u}_{bu} \cdot \boldsymbol{u}_{bv}}{\boldsymbol{u}_{bu}^2 + \boldsymbol{u}_{bv}^2} = \frac{d_b}{c_b} < 1$. When $\lambda_b$ satisfies:

$$1 > \lambda_b > \max\left(0, \frac{d_b - d_a + c_a\lambda_a}{c_b}\right), \tag{13}$$

we have

$$\lambda_b > \frac{d_b - d_a + c_a \lambda_a}{c_b}$$

$$\lambda_b c_b > d_b - d_a + c_a \lambda_a$$

$$\lambda_b c_b > d_b - (d_a - c_a \lambda_a)$$

$$\lambda_b c_b > d_b - \Delta \lambda_a$$

$$\Delta \lambda_a > d_b - \lambda_b c_b = \Delta \lambda_b$$

$$|\Delta \lambda_a| > \Delta \lambda_b$$

Next, we discuss if $\lambda_b$ always exists in the interval $0 < \lambda_b < 1$. Since $c_b - d_b = \boldsymbol{u}_{bu}^2 + \boldsymbol{u}_{bv}^2 - 2\boldsymbol{u}_{bu} \cdot \boldsymbol{u}_{bv} = (\boldsymbol{u}_{bu} + \boldsymbol{u}_{bv})^2 > 0$, $d_a = 2\boldsymbol{u}_{au} \cdot \boldsymbol{u}_{av} > 0$, $c_a = \boldsymbol{u}_{au}^2 + \boldsymbol{u}_{av}^2 > 0$, and $-1 < \lambda_a < 0$, we obtain

$$1 > \frac{d_b - d_a + c_a \lambda_a}{c_b}$$

$$c_b > d_b - d_a + c_a \lambda_a$$

$$\lambda_a < 0 \leq \frac{d_a + (c_b - d_b)}{c_a}$$

$$c_a \lambda_a < d_a + (c_b - d_b)$$

$$d_b - d_a + c_a \lambda_a < c_b$$

$$1 > \frac{d_b - d_a + c_a \lambda_a}{c_b}.$$

Therefore, the lower bound of $\lambda_b$, represented as $\max\left(0, \frac{d_b - d_a + c_a \lambda_a}{c_b}\right)$, is always smaller than the upper bound of $\lambda_b$, which indicates that there always exists $\lambda_b$ making $|\Delta \lambda_a| > \Delta \lambda_b$ when $\Delta \lambda_b > 0$. Thus, we summarize that under $\Delta \lambda_b \geq 0$ and Inequality(13), we conclude that $|\Delta \lambda_a| > \Delta \lambda_b = |\Delta \lambda_b|$.

2. For $\Delta \lambda_b < 0$, we have $\lambda_b > \frac{2\boldsymbol{u}_{bu} \cdot \boldsymbol{u}_{bv}}{\boldsymbol{u}_{bu}^2 + \boldsymbol{u}_{bv}^2} > 0$. We first consider the second condition in Inequality(4):

$$-1 < \lambda_a < \min\left(\frac{d_a - c_b + d_b}{c_a}, 0\right).$$

With this condition, we can have

$$c_a \lambda_a < d_a - c_b + d_b$$

$$\frac{d_b + d_a - c_a \lambda_a}{c_b} > 1.$$

Considering $\lambda_b$ represents the low-frequency component and $0 < \lambda_b < 1$, we can always have:

$$\lambda_b < 1 \leq \frac{d_b + d_a - c_a \lambda_a}{c_b} \tag{14}$$

$$c_b \lambda_b < d_b + d_a - c_a \lambda_a$$

$$-(d_b - c_b \lambda_b) < d_a - c_a \lambda_a$$

$$-\Delta \lambda_b < \Delta \lambda_a$$

$$|\Delta \lambda_b| < |\Delta \lambda_a|.$$

Thus, we summarize the condition under $\Delta \lambda_b < 0$ as

$$\frac{d_b}{c_b} < \lambda_b < 1. \tag{15}$$

Next, we discuss if $\lambda_a$ always exists in the interval $-1 < \lambda_a < 0$. Therefore, the lower bound of $\lambda_a$ should be smaller than the upper bound of $\lambda_a$, which indicates that the following inequality should be

satisfied

$$-1 < \frac{d_a - c_b + d_b}{c_a}$$

$$c_b - d_b < c_a + d_a$$

$$(\boldsymbol{u}_{bu} - \boldsymbol{u}_{bv})^2 < (\boldsymbol{u}_{au} + \boldsymbol{u}_{av})^2. \tag{16}$$

Here Inequality(16) is assumed in Theorem 1 as Inequality(3).

Combining the conditions Inequality(13) and Inequality(15) for two cases of $\Delta\lambda_b$ concludes the first condition in Inequality(4), which concludes the proof.

**Case 2** (Deletion). One edge is deleted from the graph, *i.e.*, $\mathcal{E} \backslash e_{uv}$. Therefore,

$$\Delta\lambda_y = -2\boldsymbol{u}_{yu} \cdot \boldsymbol{u}_{yv} + \lambda_y(\boldsymbol{u}_{yu}^2 + \boldsymbol{u}_{yv}^2). \tag{17}$$

It is the trivial extension of one-edge insertion case as the opposite number of $\Delta\lambda_y$ and leads to the same conditions. Therefore, we omit the proof here. □

## H.3 Proof of Theorem 2

*Proof.* **Theorem 2.** To make the conclusion from Theorem 1 stands for all $-1 < \lambda_a < 0$, we first conclude the condition Inequality(13) for $\Delta\lambda_b > 0$. Then, for $\lambda_b < 0$, the $|\Delta\lambda_a| > -\Delta\lambda_b = |\Delta_b|$ will hold if

$$\lambda_b < \min\left(\frac{d_b + d_a - c_a\lambda_a}{c_b}, 1\right) \tag{18}$$

according to Inequality(14) for all $-1 < \lambda_a < 0$. Then, we discuss that the upper bound of $\lambda_b$ is always greater than 0. Since $-1 < \lambda_a < 0$, then we obtain

$$-c_a\lambda_a > 0$$

$$d_b + d_a - c_a\lambda_a > d_b + d_a > 0$$

$$\frac{d_b + d_a - c_a\lambda_a}{c_b} > \frac{d_b + d_a}{c_b} > 0.$$

Therefore, there always exists $\lambda_b$ holding $|\Delta\lambda_a| > -\Delta\lambda_b = |\Delta_b|$ for $\Delta\lambda_b < 0$.

Next, we discuss the lower bound and upper bound of $\lambda_b$ for all kinds of $\Delta\lambda_b$. We derive

$$\frac{d_b + d_a - c_a\lambda_a}{c_b} - \frac{d_b - d_a + c_a\lambda_a}{c_b}$$

$$= \frac{d_b + d_a - c_a\lambda_a - d_b + d_a - c_a\lambda_a}{c_b}$$

$$= \frac{2d_a - 2c_a\lambda_a}{c_b}$$

$$> 0,$$

where $c_a > 0$, $-1 < \lambda_a < 0$, $d_a > 0$, and $c_b > 0$. Hence, we have

$$\frac{d_b + d_a - c_a\lambda_a}{c_b} > \frac{d_b - d_a + c_a\lambda_a}{c_b},$$

which indicates that the upper bound of $\lambda_b$ is always larger than the lower bound of $\lambda_b$.

□

## H.4 Proofs of Corollary 1, Corollary 2, Corollary 3 and Corollary 4

*Proof.* **Corollary 3.** Suppose every edge added/deleted from the consecutive adding/deleting process satisfies Theorem 1, we have

$$\sum_{v \in \mathcal{P}} |\Delta\lambda_{av}| > \sum_{v \in \mathcal{P}} |\Delta\lambda_{bv}| \tag{19}$$

holds for $\lambda_a$ and $\lambda_b$ satisfy

$$\max_{v \in \mathcal{P}} \Big( 0, \frac{d_{bv} - d_{av} + c_{av}\lambda_a}{c_{bv}} \Big) < \lambda_b < 1;$$

$$-1 < \lambda_a < \min_{v \in \mathcal{P}} \Big( \frac{d_{av} - c_{bv} + d_{bv}}{c_{av}}, 0 \Big),$$

where $c_{av} = \boldsymbol{u}_{au}^2 + \boldsymbol{u}_{av}^2$, $d_{av} = 2\boldsymbol{u}_{au} \cdot \boldsymbol{u}_{av}$, $c_{bv} = \boldsymbol{u}_{bu}^2 + \boldsymbol{u}_{bv}^2$, and $d_{bv} = 2\boldsymbol{u}_{bu} \cdot \boldsymbol{u}_{bv}$.

Given the non-negativity of $\boldsymbol{U}$, we have $\Delta\lambda_{av} > 0$ for all $v \in \mathcal{P}$ under edge insertion, and $\Delta\lambda_{av} < 0$ for all $v \in \mathcal{P}$ under edge deletion. Thus, we have $|\sum_{v \in \mathcal{P}} \Delta\lambda_{av}| = \sum_{v \in \mathcal{P}} |\Delta\lambda_{av}|$. While the sign of $\Delta\lambda_{bv}$ is not determined, we have $|\sum_{v \in \mathcal{P}} \Delta\lambda_{bv}| \leq \sum_{v \in \mathcal{P}} |\Delta\lambda_{bv}|$ by Cauchy Inequality. Substituting both of them in Inequality(19) concludes the proof. □

The proof for Corollary 1 is straightforward from the proof for Corollary 3, and the proofs of Corollary 2 and Corollary 4 are the trivial extension of Theorem 2 with the help of Corollary 1 and Corollary 3. Thus we omit the details here.

# I  Additional Experimental Results

## I.1  Overview of Datasets

We mainly focus on five node classification benchmark datasets under the semi-supervised setting. We use three citation networks with binary features: Cora [37], Citeseer and Pubmed [43], which aim to classify the research topics of papers, and two different types of datasets from GNN-Benchmark [44]: Coauthor CS is a coauthor network aiming to predict the most active field of study for each author from the KDD Cup 2016 challenge[4], and Amazon Photo [36] is a co-purchase network that intends to predict the category of products from Amazon. A statistical overview of all datasets is provided in Table 7.

Table 7: Dataset statistics. Edge density describes the fraction of all possible edges in the graph.

| Dataset | Nodes | Edges | Classes | Features | Edge density |
|---|---|---|---|---|---|
| **Citeseer** | 3,327 | 4,732 | 6 | 3,703 | 0.0004 |
| **Cora** | 2,708 | 5,429 | 7 | 1,433 | 0.0004 |
| **Pubmed** | 19,717 | 44,338 | 3 | 500 | 0.0001 |
| **Coauthor CS** | 18,333 | 81,894 | 15 | 6,805 | 0.0001 |
| **Amazon Photo** | 7,487 | 11,9043 | 8 | 745 | 0.0011 |

## I.2  Adversarial Samples Generation

We evaluate our model under three kinds of adversarial attack settings from the existing studies: *targeted attack with one-edge perturbation* (Nettack-One [70]), *targeted attack with multi-edge perturbation* (Nettack-Multi [70]), and *non-targeted attack* (Mettack [71]). In the targeted attack settings, following the same protocol in [70], we select 40 correctly classified nodes as targets for each dataset. The selected nodes include 10 nodes with the largest classification margin, 20 random nodes, and 10 nodes with the smallest margin. We run the whole attack and defense procedure for each target node and report average classification accuracy. Note that every successfully attacked node will contribute to a $0.025$ decrease in accuracy. To be consistent with previous works [63], we set the perturbation budget $\Delta = 1$ for the targeted node under Nettack-One and $\Delta = \mathcal{N}_u$ under Nettack-Multi. In the non-targeted attack setting, due to the high computational cost of Mettack, we set the perturbation rate as $10\%$ (*i.e.*, $\Delta = 0.1E$) with the 'A-Meta-Self' training strategy.

## I.3  Experimental Setups

To select hyperparameters and GCN model architectures, we closely follow guidelines of original authors and relevant papers on GCNs (VanillaGCN [27], JK-Net [61], SGC [54], Graph-U-Net [16], and

---

[4]https://kddcup2016.azurewebsites.net

GCNII [10]), baseline defense algorithms (GCN-Jaccard [55], GCN-SVD [14], GNNGUARD [63], and Pro-GNN [25]), and models for generating adversarial attacks (Nettack-One [70], Nettack-Multi [70], and Mettack [71]). Note that our motivation is to provide a robust training paradigm that enhances the robustness of any given GCN backbone, thus some recent efforts on proposing new variants of GCNs are not eligible for comparison.

Across all experiments, for a fair comparison, we keep the common hyperparameters as the same, including the number of layers as $2$, hidden units as $128$, learning rate as $0.01$, weight decay as $5e-4$, the number of epochs as $200$ and dropout rate as $0.5$. The cross-entropy loss is optimized using the Adam optimizer [26]. For GCN-LFR, we set the ratio of top low-frequency to $k = d\%N$, which indicates that we treat the top $d\%$ smallest eigenvalues as low-frequency components. For each dataset, $d$ is chosen from grid search in $\{5, 10, \cdots, 50\}$. Meanwhile, we set the weight coefficient $\alpha$ for balancing two losses to $0.5$, unless otherwise stated.

We utilize a PyTorch based package, DeepRobust (`https://github.com/DSE-MSU/DeepRobust`) [34], to implement the adversarial attack models and baseline defense algorithms except for GNNGUARD, which is from their public code. We utilize the PyTorch Geometric package (`https://github.com/rusty1s/pytorch_geometric`) to acquire all mentioned datasets and implement GCN models. The training and evaluation of models are realized with PyTorch (`https://pytorch.org`). For all methods, we report the performance of single run for Nettack-One, and repeat every experiment $5$ times then report the mean performance for Nettack-Multi and Mettack. For other parameters, we follow the setup in [70, 71]. We have run most of the experiments on a single 16GB GeForce GTX TITAN X GPU.

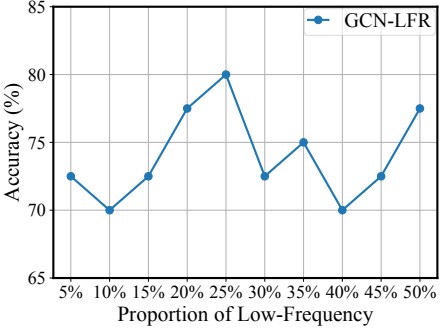

Figure 6: Grid search on the ratio of low-frequency.

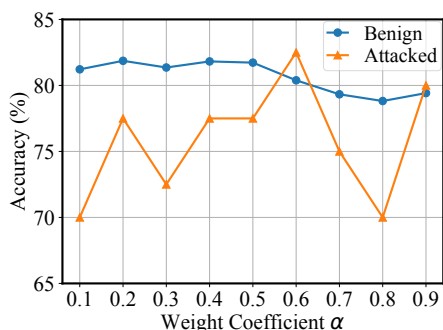

Figure 7: Grid search on the weight coefficient $\alpha$.

### I.4 T-test on the Performance of GCN-LFR on Clean Datasets.

For better demonstration of Table 4, we perform significance tests to verify that GCN-LFR achieves on par performance in comparison with the backbone using double-sided T-test. We use Python package `scipy.stats.ttest_ind` and report the average results over different backbones in Table 8. We can find that most of the p-value is larger than $0.05$, thus we cannot reject the null hypothesis of identical average scores, which supports our claim.

Table 8: T-test on the classification accuracy over all benign datasets (*i.e.*, non-attacked) w or w/o LFR.

| Dataset | Cora | Citeseer | Pubmed | Coauthor CS | Amazon Photo |
|---------|------|----------|--------|-------------|--------------|
| p-value | 0.610 | 0.198 | 0.313 | 0.005 | 0.190 |

### I.5 Sensitivity Analysis on the Ratio of Low-frequency and Weight Coefficient $\alpha$

We choose to set the weight coefficient $\alpha$ as $0.5$ simply because this setting has already shown consistently better performance across all datasets. Thus we choose not to tune this hyperparameter. However, for better illustration, we provide the defending results against one-edge perturbation on Cora dataset upon the grid search on the proportion of low-frequency within $\{5\%, 10\%, \cdots, 50\%\}$ in

Figure 6. Interestingly, we can find that the robustness of GCN-LFR stops enhancing when the ratio becomes large. Meanwhile, we also conduct the sensitivity analysis of GCN-LFR w.r.t $\alpha$ on Cora dataset under both benign (no attack happens and tested on the full testset) and one-edge perturbation (tested on the selected 40 nodes as targets) within $\{0.1, 0.2, \cdots, 0.9\}$ in Figure 7. We can observe that GCN-LFR is relatively insensitive to $\alpha$ and stays effective even under extreme conditions. Relatively larger $\alpha$ also leads to better robustness but worse performance in benign situation.

## I.6 Parameter Sharing Architecture

The two-channel cooperation with parameter sharing aims to transfer the robustness of GCN-LFR to the main channel through regularization. To further demonstrate the effectiveness of this architecture, we implement a variant of GCN-LFR but only without parameter sharing, which we denote as GCN-LFR-NoShare. Table 9 demonstrates the defense performance of three models by reporting the accuracy after attack. We can observe that even we do not share the parameters between the main channel and the LFR channel, the defense performance can still be enhanced.

Table 9: Defense performance (multi-class classification accuracy) against non-targeted attacks under Mettack.

| Dataset | VanillaGCN | GCN-LFR-NoShare | GCN-LFR |
|---|---|---|---|
| Cora | $0.771 \pm 0.010$ | $0.778 \pm 0.012$ | $\mathbf{0.791 \pm 0.012}$ |
| Citeseer | $0.654 \pm 0.012$ | $0.663 \pm 0.012$ | $\mathbf{0.670 \pm 0.015}$ |
| Amazon Photo | $0.824 \pm 0.012$ | $0.670 \pm 0.015$ | $\mathbf{0.843 \pm 0.015}$ |

## I.7 Running Time for Low-Frequency Components

The extra complexity brought by our proposed method only lies in the acquisition of the low-frequency components. Thus aside from the complexity analysis, we provide the mean time of this procedure averaging from 5 runs on all datasets in Table 10. We can find that this burden resulting from LFR is acceptable across all datasets.

Table 10: Running time for preprocessing of low-frequency components.

| Dataset | Cora | Citeseer | Pubmed | Coauthor CS | Amazon Photo |
|---|---|---|---|---|---|
| LFR Burden (s) | 2.351 | 3.764 | 480.583 | 426.237 | 55.003 |

Table 11: Classification accuracy on both benign and attacked heterophily datasets. The first row reports the results of VanillaGCN. The second row reports the results of JK-Net.

| Dataset | VanillaGCN | | GNNGUARD VanillaGCN | | GCN-LFR VanillaGCN | |
|---|---|---|---|---|---|---|
| | Benign | Attacked | Benign | Attacked | Benign | Attacked |
| Wisconsin | $0.478 \pm 0.013$ | $0.434 \pm 0.067$ | $0.487 \pm 0.016$ | $0.461 \pm 0.063$ | $\mathbf{0.526 \pm 0.019}$ | $\mathbf{0.480 \pm 0.033}$ |
| Chameleon | $0.496 \pm 0.014$ | $0.395 \pm 0.012$ | $0.488 \pm 0.011$ | $0.472 \pm 0.010$ | $\mathbf{0.516 \pm 0.032}$ | $\mathbf{0.446 \pm 0.021}$ |
| | JK-Net | | GNNGUARD JK-Net | | GCN-LFR JK-Net | |
| Wisconsin | $0.488 \pm 0.014$ | $0.433 \pm 0.023$ | $0.509 \pm 0.022$ | $0.469 \pm 0.031$ | $\mathbf{0.523 \pm 0.058}$ | $\mathbf{0.483 \pm 0.025}$ |
| Chameleon | $0.494 \pm 0.013$ | $0.386 \pm 0.027$ | $0.461 \pm 0.012$ | $0.441 \pm 0.021$ | $\mathbf{0.553 \pm 0.039}$ | $\mathbf{0.465 \pm 0.026}$ |

## I.8 Defense for Heterophily Datasets

The study considering both low and high frequencies in GNNs grows rapidly in recent years [33, 65]. Current research suggests that GCNs are implicit low-pass filtering models [54, 35]. Thus the high-frequency components, which are mostly filtered out during training, would have less impact on the final learnt representations. Meanwhile, the high-frequency components may contain more noise in comparison with low-frequency ones even on the clean graphs [35], which further limits their contribution to the robustness of GCNs. Thus in this paper, we focus on studying the robustness property for the low-frequency components and prove that not all low-frequency components are robust against adversarial attacks.

For better demonstration, we also choose two representative heterophily datasets, *i.e.*, Wisconsin and Chameleon [40, 68], and evaluate the performance under both benign and adversarial settings (with Mettack). The results can be found in Table 11. One can find that our proposed method still enhances the performance of GCN backbone on both clean and poisoned graphs. This is consistent with our theoretical findings since we do not impose assumptions on the homophily of the underlying datasets. The investigation on high-frequency components might lead to a promising research direction and we leave it as interesting future work.