# OpenReview forum: "Not All Low-Pass Filters are Robust in Graph Convolutional Networks"
_NeurIPS.cc/2021/Conference — NeurIPS 2021 Poster_

### Official Review · Reviewer_iiUG · 2021-07-13

**Rating:** 6
**Confidence:** 4

**Summary:**

The authors introduce a method to improve the robustness of graph neural networks (GNNs). Specifically, the method performs robust training to defend against structure poisoning attacks. The algorithm learns low-pass filter coefficients $\mathbf{F}$. A GCN which uses this low-pass filter as the graph convolution matrix shares weights with a standard GCN during training. The authors compare results on a set of established datasets and attacks.

**Limitations And Societal Impact:**

There is no broader impact statement in the main text of the paper. While the authors have put it into the appendix, this practice is not fair to other authors who sacrifice precious space in the main text of the paper to have the broader impact statement, as demanded by the NeurIPS guidelines.


**Main Review:**

Strengths:
* The authors report strong robustness results across datasets and attacks.
* The theoretical analysis is insightful.
* The presentation of the paper is good and clear.

Weaknesses:
* The ablation study is weak and does not sufficiently support the claim that learning the filter weights $\mathbf{F}$ is better than the model where the eigenvalues are used (more in detailed comments).
* The complexity analysis is misleading, showing $O(ckN)$ where it is actually $O(cN^2)$ (more in detailed comments).
* The clean accuracy of the method is quite consistently lower than the original accuracy.

Detailed comments:

* Since the graph convolution matrix multiplication for M_LFR is no longer sparse, what effect does this have on the scalability of the method?
* In the Appendix, the authors state that the number $k$ of low frequency components considered is $0.25 N$. With this, the complexity to compute the top-$k$ eigenpairs is $O(cN^2)$. This is a major scalability limitation and should be addressed in the main text.
* Are predictions of the low frequency GCN used at inference time? If yes, this means that the method cannot be used in an inductive fashion on new nodes at inference time.
* It seems like the method is also well suitable for evasion attacks. How does the method defend against these?
* Figure 3 (right) does not support the claim that "GCN-LFR achieves substantially better validation accuracy than that of GCN-LFRFix". The scale is too coarse to see much difference and there is quite some fluctuation in the curves.
* Figure 3 (right) also does not support the claim that "GCN-LFR achieves competitive performance against GCN", since it only shows the validation accuracy. Also, this is just a single run on one dataset.
* How does GCN-LRF_fix compare to GCN-LRF in a proper tabular comparison on multiple datasets? This is crucial to determine the usefulness learning the filter over keeping it fixed, and Figure 3 does not answer this as stated above.
* Similarly, Figure 4 (left) is not suited to draw general conclusions such as "GCN-LFR_Alter achieves better training and validation performance compared with GCN-LFRJoint". As above, please run a proper comparison on multiple splits and datasets.
* The results in Table 1 for GNNGuard seem to be substantially weaker than reported in Table 1 of the original paper. For example, for Citeseer they report 72.0% robustness after attack and 72.1% clean accuracy. This would correspond to an accurracy of 99.86% in the scale reported by the authors. What is the reason for this?
* Table 4 seems to suggest that GCN-LFR has lower accuracy on average. On how many splits are these results computed? What are the confidence intervals?

**Time Spent Reviewing:**

5

---

> ### Author Response · Authors · 2021-08-10
> **Why broader impact in the Appendix**
>
> > There is no broader impact statement in the main text of the paper. While the authors have put it into the appendix, this practice is not fair to other authors who sacrifice precious space in the main text of the paper to have the broader impact statement, as demanded by the NeurIPS guidelines.
>
> Thank you for pointing this out. We strictly follow the FAQ of NeurIPS guidelines (https://neurips.cc/Conferences/2021/PaperInformation/NeurIPS-FAQ), under the Question `Is a broader impact statement required and must it fit within the page limit?`, there is a specific instruction as we quote here:
> > "...You may include a discussion of these potential negative societal impacts anywhere in the paper (in the intro, in the conclusion, as a stand-alone section, in the supplemental material if appropriate, etc.)...".
>
> Therefore, it is appropriate to have the broader impact statement in the supplementary material as we did.

---

> ### Author Response · Authors · 2021-08-10
> **Concerns upon Table 4**
>
> > Table 4 seems to suggest that GCN-LFR has lower accuracy on average. On how many splits are these results computed? What are the confidence intervals?
>
> In Table 4, we repeated every experiment 5 times, which indicates five splits totally, and report average performance across independent runs. This is the identical setting from GNNGUARD, which can be referred to Appendix E of their original paper. Furthermore, as shown in the previous table, the standard deviations of our method are quite small (around 0.020), and the performance of our model is very stable across different splits. Thus we omit the stds due to the space limit for the large tables. Our experimental findings reveal that GCN-LFR could achieve on par performance on clean datasets and enhance the robustness of the given model by a large margin.
> For a better demonstration of Table 4, we perform significance tests to verify that GCN-LFR achieves on par performance in comparison with the backbone using double-sided T-test. We use Python package "scipy.stats.ttest_ind" and report the average results over different backbones as follows:
>
> |     | Cora | Citeseer | Pubmed | Coauthor CS | Amazon Photo |
> |---------|-------|----------|--------|-------------|--------------|
> | p-value | 0.610 | 0.198  | 0.313 | 0.005    | 0.190    |
>
> We can find that most of the p-values are larger than 0.05. Thus we cannot reject the null hypothesis of identical average scores, which supports our claim.

---

> ### Author Response · Authors · 2021-08-10
> **The performance of GNNGUARD and the metric we used for evaluating the robustness**
>
> > The results in Table 1 for GNNGuard seem to be substantially weaker than reported in Table 1 of the original paper. For example, for Citeseer they report 72.0% robustness after attack and 72.1% clean accuracy. This would correspond to an accurracy of 99.86% in the scale reported by the authors. What is the reason for this?
>
> Thank you for pointing this out, and this is quite a vague description from the original GNNGUARD paper that easily leads to missing understanding. The column for No Attack in its original paper indicates the test accuracy on the **whole test set**, which corresponds to the 20% share of splits. While the column of defense accuracy from their original paper indicates the test accuracy after the attack on the **selected 40 nodes**, which is the same setting as ours. The test accuracy of this set is 100% since only the correctly classified results are chosen as candidates of 40 nodes here, and every successfully attacked node will contribute to a 0.025 decrease in accuracy, which we declare from Line 715-716 in our supplementary material.
>
> Specifically, GNNGUARD set the budget of target attack as the degree of target nodes, which corresponds to Nettack-Multi in our setting. In their official implementation, they skipped the selected nodes with perturbation of less than one edge, which results in the candidate set could be less than 40 nodes. This is the reason that their results are not multiple of 0.025. On the contrary, we choose to maintain the 40 selected nodes as candidates for consistent demonstration, and this could be the reason for the substantially weaker results of GNNGUARD. Many thanks again for catching this, and we will make this clearer in our revision.

---

> ### Author Response · Authors · 2021-08-10
> **Concerns upon the insufficient demonstration from Figure 4**
>
> > Similarly, Figure 4 (left) is not suited to draw general conclusions such as "GCN-LFR_Alter achieves better training and validation performance compared with GCN-LFRJoint". As above, please run a proper comparison on multiple splits and datasets.
>
> Many thanks again for this valuable suggestion as to the ones for Figure 3. Due to the limitation of the response. We provide the train/val/test accuracies across all datasets with multiple runs. Specifically, We run the experiments with five different random seeds, which indicate different data splits, and report mean and standard deviation in the following table:
>
> |           | Cora     | Citeseer   | Pubmed    | Coauthor CS  | Amazon Photo |
> |---------------------|---------------|---------------|---------------|---------------|---------------|
> | GCN-LFR-Joint-Train | 0.999 ± 0.001 | 0.991 ± 0.004 | 0.929 ± 0.002 | 0.986 ± 0.004 | 0.975 ± 0.007 |
> | GCN-LFR-Alter-Train | 0.999 ± 0.001 | 0.995 ± 0.002 | 0.932 ± 0.013 | 0.977 ± 0.006 | 0.986 ± 0.005 |
> | GCN-LFR-Joint-Val  | 0.827 ± 0.018 | 0.702 ± 0.027 | 0.832 ± 0.006 | 0.903 ± 0.008 | 0.939 ± 0.004 |
> | GCN-LFR-Alter-Val  | 0.837 ± 0.025 | 0.717 ± 0.008 | 0.843 ± 0.008 | 0.907 ± 0.007 | 0.942 ± 0.009 |
> | GCN-LFR-Joint-Test | 0.817 ± 0.011 | 0.698 ± 0.007 | 0.846 ± 0.005 | 0.908 ± 0.003 | 0.917 ± 0.001 |
> | GCN-LFR-Alter-Test | 0.833 ± 0.009 | 0.701 ± 0.011 | 0.853 ± 0.010 | 0.912 ± 0.008 | 0.931 ± 0.015 |
>
> From this table, we can draw the consistent conclusion that we claim in the analysis of Section 5.3. We will update Figure 4 with more datasets and multiple runs in the revised version.

---

> ### Author Response · Authors · 2021-08-10
> **Concerns upon Figure 3 and tabular comparison between GCN_LFR_Fix and GCN_LFR**
>
> > Figure 3 (right) does not support the claim that "GCN-LFR achieves substantially better validation accuracy than that of GCN-LFRFix". The scale is too coarse to see much difference and there is quite some fluctuation in the curves.
> > Figure 3 (right) also does not support the claim that "GCN-LFR achieves competitive performance against GCN", since it only shows the validation accuracy. Also, this is just a single run on one dataset.
> > How does GCN-LRF_fix compare to GCN-LRF in a proper tabular comparison on multiple datasets? This is crucial to determine the usefulness learning the filter over keeping it fixed, and Figure 3 does not answer this as stated above.
>
> We consider these concerns as that Figure 3 is insufficient for a general claim for the relationship among GCN, GCN-LFR-Fix, and GCN-LFR, and a proper tabular comparison on multiple datasets with multiple splits is suggested. This is an excellent suggestion, and we completely agree with the reviewer on this since Figure 3 is insufficient to provide confident support for our claim. Hence, we provide the full table here under both benign and attacked settings for better demonstration:
>
> |               | Cora            | Citeseer          | Pubmed           | Coauthor CS         | Amazon Photo        |
> |------------------------------|-----------------------------|-----------------------------|-----------------------------|-----------------------------|-----------------------------|
> | GCN (clean/attacked)     | 0.830 ± 0.011/0.771 ± 0.010 | 0.720 ± 0.012/0.654 ± 0.012 | 0.860 ± 0.018/0.794 ± 0.013 | 0.902 ± 0.004/0.816 ± 0.006 | 0.929 ± 0.002/0.824 ± 0.004 |
> | GCN-LFR-Fix (clean/attacked) | 0.779 ± 0.005/0.726 ± 0.012 | 0.683 ± 0.013/0.634 ± 0.015 | 0.832 ± 0.013/0.786 ± 0.015 | 0.883 ± 0.006/0.808 ± 0.005 | 0.876 ± 0.003/0.793 ± 0.015 |
> | GCN-LFR (clean/attacked)   | 0.824 ± 0.005/0.791 ± 0.012 | 0.706 ± 0.013/0.670 ± 0.015 | 0.856 ± 0.015/0.802 ± 0.012 | 0.912 ± 0.003/0.837 ± 0.015 | 0.928 ± 0.003/0.843 ± 0.015 |
>
> Combining the results from Figure 5, we can find that `GCN-LFR` has both better performance and robustness over `GCN` and` GCN-LFR-Fix`. Interestingly, the better generalization ability of `GCN-LFR` over `GCN-LFR-Fix` could be observed implicitly either in this way. We will replace Figure 3 with this clear tabular demonstration in the revised version as suggested.

---

> ### Author Response · Authors · 2021-08-10
> **Defense against evasion attacks**
>
> > It seems like the method is also well suitable for evasion attacks. How does the method defend against these?
>
> Thank you for your interest. Actually, we have conducted the evasion attack experiments and found the performance deduction of our proposed method. This is because that our `GCN-LFR` model needs to enhance the robustness of a given GCN during regularized retraining on the attacked graph. Without retraining, the Auxiliary Regularization Net $M_{LFR}$ can not transfer the robustness to the main model $M_{GCN}$. Therefore, even though our theoretical analysis is not limited to the specific attacking setting, the current method design is more suitable for defending the poisoning attack. How to design a method that can handle the evasion attacks is an interesting further work.

---

> ### Author Response · Authors · 2021-08-10
> **Can our method be used in the inductive setting?**
>
> > Are predictions of the low frequency GCN used at inference time? If yes, this means that the method cannot be used in an inductive fashion on new nodes at inference time.
>
> Yes, our method can be adopted to inductive settings, since during the inference stage, we only need the main model $M_{GCN}$ to perform the inference. It's possible to apply the trained model to predict the new nodes. However, according to our theoretical analysis, $M_{LFR}$ can only extract the robustness information from the input graph and transfer it to $M_{GCN}$ by parameter-sharing. Even though we can apply a trained model to a new graph in an inductive fashion, the behavior and robustness of the model are beyond the scope of our theoretical analysis.

---

> ### Author Response · Authors · 2021-08-10
> **Inaccurate claim for the extra complexity $O(ckN)$**
>
>
> > In the Appendix, the authors state that the number $k$ of low-frequency components considered is $0.25N$. With this, the complexity to compute the top-$k$ eigenpairs is $\mathcal{O}(cN^2)$. This is a major scalability limitation and should be addressed in the main text.
>
> Many thanks for catching this inaccurate claim. In our analysis, we assume that the fixed top-$k$ eigenpairs are utilized, such as top-$10$ eigenpairs across all datasets. Thus we state that the extra complexity $O(ckN)$. It is true that during implementation, we would like to make our method more adaptive to different graphs, then we choose to use a proportion of the eigenvalues, which would result in $O(cN^{2})$ extra complexity. We are sorry for this inaccurate statement and will clarify it in the main text of the revised version as suggested.
>
> Meanwhile, we would like to mention that from our observation from Appendix Section G.5. We can find that the robustness of `GCN-LFR` is still enhanced by a great margin than Vanilla GCN when $k = 0.05N$, which implies that use a fixed and small top-$k$ eigenpair could also achieve the enhancement in practice.

---

> ### Author Response · Authors · 2021-08-10
> **No longer sparse graph convolution matrix $M_{LFR}$**
>
>
> > Since the graph convolution matrix multiplication for M_LFR is no longer sparse, what effect does this have on the scalability of the method?
>
> Thank you for this insightful question! Actually, for small graphs, the increase of consuming time from matrix multiplication between dense matrices is nearly unnoticeable. While for the large graphs, we choose to maintain the sparseness of the graph convolution matrix by keeping the eigenvectors $U$ sparse with a threshold $0.0001$, which can be referred to our submitted code (Line 152 from LFR_attack_net.py). In this way, the scalability of the method can be consistent with our complexity analysis.
>
> In fact, this is a common issue for the spectral GNNs if reconstruction is needed. For example, the complexity of Vanilla GCN would be $\mathcal{O}(N^2M + NM^2)$ with dense matrix multiplication rather than $\mathcal{O}(EM + NM^2)$ which is with sparse matrix multiplication, where the worse case of the edges on a graph $E$ is $\mathcal{O}(N^2)$. This also reveals that the main burden would come from the dense matrix multiplication since it is of the same order as the extra complexity brought by our proposed method. Considering our main contribution is on the theoretical analysis and the scalability is not our main focus, the extra complexity could be acceptable. We would make the analysis of complexity clearer in the revised version.

---

> > ### Comment · Reviewer_iiUG · 2021-08-21
> > **Reviewer note**
> >
> > Thank you for your thoughtful and thorough replies. My concerns were addressed. In particular, I find the new results table of LFR-Fix vs LFR interesting and important to include in the updated main text. I have increased my score accordingly.

---

> > > ### Author Response · Authors · 2021-08-23
> > > **Many Thanks to Your Response and Support**
> > >
> > > We really appreciate your thoughtful and constructive response to our feedback, and we are glad that your concerns have been addressed.
> > >
> > > In reflection of your suggestions, we will include the corresponding improvements in the revised version, especially updating the new results regarding LFR-Fix vs LFR to the main text.
> > >
> > > Would you please let us know if you have any further recommendations for how we can enhance our submission?

---

### Official Review · Reviewer_CV31 · 2021-07-16

**Rating:** 6
**Confidence:** 3

**Summary:**

Graph convolutional networks are volunerable to adversarial attacks.  As commonly recognized by researchers,  low-frequency components of GCNs are more robust than high-frequency parts of GCNs. However, the relationship between graph spectrum and robustness of GCN is not well studied. This paper theoretically proposes the robust interval of low-frequency components and regularizes the low-frequency components of GCNS within the robust interval through an auxiliary neural network. In this way,  the robustness of a GCN is enhanced.

**Ethics Review Area:**

["I don’t know"]

**Limitations And Societal Impact:**

Yes

**Main Review:**

Originality:  this paper bridges the gap between the spectrum analysis and the robustness of GCNs under structural perturbations. The depth of this paper is satisfied. The authors not only studied one-edge perturbation but also considered the multi-edge non-targeted perturbation.

Quality: the proposed method is technically sound. The experimental results support the proposed method. Its performance is evidently higher than baseline method. However, the chosen datasets are limited to homophily graph datasets. For these datasets, low-frequency components are more useful than high-frequency components. How well the proposed method suit for non-homophily datasets [*] should be further studied.
[*] New Benchmarks for Learning on Non-Homophilous Graphs

Clarity: the connection between the proposed robust frequency interval and the proposed method is weak and not well clarified. The authors should clarify whether the parameters of F are constrained within the robust interval during training.
Besides, the authors should clarify the inference procedure of the proposed framework.

Significance: this paper is a first vulnerability study between the graph spectrum and the robustness behavior of GCNs. It theoretically shows that not all low-pass filters are robust in graph convolutional networks.


**Time Spent Reviewing:**

3 hours

---

> ### Author Response · Authors · 2021-08-10
> **Robust interval and inference procedure (Clarity)**
>
> Thank you for pointing this out and we appreciate the thoughtful and constructive feedback towards clarity. We would like to address the two specific comments as follows:
>
> > The authors should clarify whether the parameters of F are constrained within the robust interval during training.
>
> 1. We think the reviewer might miss the ablation study that checks how many parameters of $F$ fall into the robust interval during training, which is the last paragraph of Section 5.3, from Line 348 to 358. As our explanation from Section 4.3.1, explicitly computing the robustness interval is infeasible before the structural attack since it depends on which edge is perturbed. Thus we alleviate this obstacle by proposing a set of learnable parameters $F$ to capture the robust interval. The results from our empirical check confirm the effectiveness of this design.
>
> > Besides, the authors should clarify the inference procedure of the proposed framework.
>
> 2. As for the inference procedure, we are sorry that this part is a little vague from our description. Actually, as a method of regularization, we only transfer the robustness of low-frequency components by co-training an auxiliary net $M_{LFR}$ and sharing its parameters during the training procedure, and we only use the prediction of the given model $M_{GCN}$ for inference. In other words, the extra complexity of our proposed method only from the training procedure, while the inference procedure stays the same, which further reflects the scalability of our proposed approach. We will add the discussion above and make our description clearer.

---

> ### Author Response · Authors · 2021-08-10
> **Experiments on heterophily datasets (Quality)**
>
> > How well the proposed method suit for non-homophily datasets should be further studied. New Benchmarks for Learning on Non-Homophilous Graphs
>
> Thank you for your interest, and this is an excellent investigative question. For better demonstration, we choose two representative heterophily datasets, i.e., Wisconsin and Chameleon[1,2], and evaluate the performance under both benign and adversarial settings. The results can be found below:
>
> | Dataset   | GCN (clean/attacked)        | GNNGuard-GCN (clean/attacked)   | GCN-LFR (clean/attacked)    | JKNet (clean/attacked)      | GNNGuard-JKNet (clean/attacked)   | JKNet-LFR (clean/attacked)  |
> |-----------|-----------------------------|-----------------------------|-----------------------------|-----------------------------|-----------------------------|-----------------------------|
> | Wisconsin | 0.478 ± 0.013/0.434 ± 0.067 | 0.487 ± 0.016/0.461 ± 0.063 | **0.526 ± 0.019/0.480 ± 0.033** | 0.488 ± 0.014/0.433 ± 0.023 | 0.509 ± 0.022/0.469 ± 0.031 | **0.526 ± 0.058/0.480 ± 0.025** |
> | Chameleon | 0.496 ± 0.014/0.395 ± 0.012 | 0.488 ± 0.011/0.472 ± 0.010 | **0.516 ± 0.032/0.446 ± 0.021** | 0.494 ± 0.013/0.386 ± 0.027 | 0.461 ± 0.012/0.441 ± 0.021 | **0.553 ± 0.039/0.465 ± 0.026** |
>
> We can find that our proposed method still enhances the performance of GCN backbone on both clean and poisoned graphs. This is consistent with our theoretical findings since we do not have assumptions on the homophily of underlying datasets. We completely agree that the investigation on high-frequency components also has merits and leads to a promising research direction, and further discussions can be referred to our response to Reviewer mqEy.
>
> Besides the aforementioned heterophily datasets that we tested, we find the size of datasets provided in the recommended paper[3] might also be too large for current attack methods to handle, with the reasons that you can refer to our response to Reviewer mqEy. We would like to leave the adaptation of our framework to these datasets as a future and add discussions accordingly in our revised version.
>
> [1] Geom-GCN: Geometric Graph Convolutional Networks, ICLR 2020
>
> [2] Beyond Homophily in Graph Neural Networks: Current Limitations and Effective Designs, NeurIPS 2020
>
> [3] New Benchmarks for Learning on Non-Homophilous Graphs, WWW 2021

---

> ### Author Response · Authors · 2021-08-26
> **Kind reminder to the reviewer**
>
> Dear Reviewer,
>
> We are wondering whether your concerns have been properly addressed.
>
> If you have further questions after reading the answers, it would be great to let us know.
>
> Best regards,
>
> The Authors

---

### Official Review · Reviewer_mqEy · 2021-07-17

**Rating:** 6
**Confidence:** 4

**Summary:**

This work tries to study the robustness of GCNs under structural perturbations from the perspective of graph spectrum, which for me is interesting. In theory, they proved that the low-frequency components can be more robust against structural perturbations if their eigenvalues fall into a robust interval. For algorithm design, they developed a regularized learning framework that combines a regular GCN-based objective and a low-frequency regularization term, named GCN-LFR. Finally, the authors tested the capability of GCN-LFR in resisting adversarial attacks on several benchmark datasets.

**Limitations And Societal Impact:**

Yes, the authors mentioned the limitations in the conclusion and societal impacts in the Appendix.

**Main Review:**

Strengths:

The idea to bridge the gap between the spectral analysis and the robustness of GCNs under structural perturbations is interesting.

Weakness:

(1) The reason why alternative training is better than joint training is not clearly explained. I am curious about the influence of parameter sharing used for two channels (GCN and LFR).

(2) It is not clear to me whether the enhanced robustness shown in experiments is caused by the model’s two-channel cooperation architecture or exactly the functionality of LFR. The threshold value to select the top-k low-frequency eigenvectors is of high importance.

(3) How is the scalability of the proposed method? Although authors provided a complexity analysis of GCN-LFR, more empirical verifications on one/two large-scale graph datasets are required. My initial impression of LFR is that it may need a huge computation burden.

Major points:

(1) In Figure 1, the authors concluded that “low-frequency components (far left) tend to be more robust in comparison with high-frequency ones (far right)”, which seems to be the case, intuitively. However, I do not feel confident about this conclusion since the ‘rate of change’ (over eigenvalues before & after perturbation), which can be more suitable and fair, is not clearly illustrated.

(2) I am curious about one issue: Do the high-frequency components also play a role in affecting the robustness of GCNs? Some discussions or empirical studies could help readers better understand the key ingredients towards how to investigate GCN’s robustness analysis from a spectrum way.

(3) More experimental verifications on large-scale graph benchmark datasets, for example, OGB, are required to further demonstrate the potential of the proposed method.


**Time Spent Reviewing:**

10

---

> ### Author Response · Authors · 2021-08-10
> **The role of high-frequency components in the robustness of GCNs (Major Points (2))**
>
> > I am curious about one issue: Do the high-frequency components also play a role in affecting the robustness of GCNs? Some discussions or empirical studies could help readers better understand the key ingredients towards how to investigate GCN’s robustness analysis from a spectrum way.
>
> This is also an excellent and inspiring question! In this paper, we focus on studying the robustness property for the low-frequency components and prove that not all low-frequency components are robust against adversarial attacks. We would like to share our  thoughts on the role that high-frequency components play in the robustness of GCNs are as follows:
>
> 1. Current research suggests that the GCNs are implicit low-pass filtering models[1,2]. Thus the high-frequency components, which are filtered out during training, would have less impact on the final representation learning. Meanwhile, the high-frequency components may contain more noise in comparison with low-frequency ones even on the clean graphs [2], which further limits their contribution to the robustness of GCNs.
>
> 2. Meanwhile, we believe that the high-frequency components could contribute to the robustness under certain situations, such as when the underlying graphs are more heterophily, which can also be referred to the evaluation of LFR on non-homophily graphs in our response to **Reviewer CV31** as we repeated here:
>
> | Dataset   | GCN (clean/attacked)        | GNNGuard-GCN (clean/attacked)   | GCN-LFR (clean/attacked)    | JKNet (clean/attacked)      | GNNGuard-JKNet (clean/attacked)   | JKNet-LFR (clean/attacked)  |
> |-----------|-----------------------------|-----------------------------|-----------------------------|-----------------------------|-----------------------------|-----------------------------|
> | Wisconsin | 0.478 ± 0.013/0.434 ± 0.067 | 0.487 ± 0.016/0.461 ± 0.063 | **0.526 ± 0.019/0.480 ± 0.033** | 0.488 ± 0.014/0.433 ± 0.023 | 0.509 ± 0.022/0.469 ± 0.031 | **0.526 ± 0.058/0.480 ± 0.025** |
> | Chameleon | 0.496 ± 0.014/0.395 ± 0.012 | 0.488 ± 0.011/0.472 ± 0.010 | **0.516 ± 0.032/0.446 ± 0.021** | 0.494 ± 0.013/0.386 ± 0.027 | 0.461 ± 0.012/0.441 ± 0.021 | **0.553 ± 0.039/0.465 ± 0.026** |
>
> &ensp; &ensp; &ensp; Our theoretical analysis also suggests that there could also be a robust interval for the high-frequencies. Considering the recent research progress on the contribution of high-frequency components on the generalization of CNNs[3], a similar pattern of high-frequency on graphs and GNNs might also exist, but the research towards this topic is still inadequate currently.
>
>
>
> In conclusion, we strongly agree that the robustness analysis of the high-frequency components is an inspiring direction. We will add the discussions above in the revised version.
>
> [1] Simplifying graph convolutional networks, ICML 2019
>
> [2] Revisiting Graph Neural Networks: All We Have is Low-Pass Filters, preprint
>
> [3] High-frequency component helps explain the generalization of convolutional neural networks, CVPR 2020

---

> > ### Comment · Reviewer_mqEy · 2021-08-28
> > **response to authors**
> >
> > Yes, I agree with your explanation and the further added empirical results are acceptable and sound logical. I also suggest the following Refs. which helps readers better follow the line of research that focuses on both low&high frequencies in GNNs.
> > [1] https://arxiv.org/abs/2102.06986
> > [2]Li, M., Ma, Z., Wang, Y. G., & Zhuang, X. (2020). Fast Haar transforms for graph neural networks. Neural Networks, 128, 188-198.

---

> > > ### Author Response · Authors · 2021-08-29
> > > **Thank you**
> > >
> > > Many thanks to your suggestion on the refs. We are glad to know these interesting works and will surely add them to the discussion towards the current efforts on analyzing both low&high frequencies in GNNs in the updated version.

---

> ### Author Response · Authors · 2021-08-10
> **The illustration of the significant change on the spectrum (Major Points (1))**
>
> > However, I do not feel confident about this conclusion since the ‘rate of change’ (over eigenvalues before & after perturbation), which can be more suitable and fair, is not clearly illustrated.
>
> This is a great point! We agree that there are two ways to define the significant change in eigenvalues. One is the `absolute change` as we demonstrated in  Figure 1, and the other one is the suggested `rate of change`.  We choose `absolute change` since it is more suitable to analyze the model's robustness. Here is our detailed analysis:
>
> > Considering the perturbation on the all frequencies as a diagonal matrix $\epsilon$, where $\epsilon_{ii}$ is the absolute change on $i_{th}$ eigenvalue. Thus for the original adjacency matrix $A = U \Lambda U^{T}$, the reconstructed adjacency matrix after perturbation by $\epsilon$ is approximated as $A' = U (\Lambda + \epsilon) U^{T} = U \Lambda U^{T} + U \epsilon U^{T}$. Thus the absolute change can better reflect the perturbation on the reconstructed $A'$, which is directly related to the analysis of robustness.
>
> In summary,  the `absolute change`  is suitable and fair for the analysis on the behavior of robustness of GNNs under our framework. Meanwhile, we thank you for pointing this out, and we will add this 'rate of change' metric on Figure 1 for better demonstration, along with the discussion above in the revised version.

---

> > ### Comment · Reviewer_mqEy · 2021-08-28
> > **response to authors**
> >
> > Sure, thank you for clarifying this point.

---

> ### Author Response · Authors · 2021-08-10
> **The scalability of the proposed framework (Weakness (3) and Major points (3))**
>
> Thank you for pointing this important question out and your acknowledgment of our complexity analysis, our responses to this concern are from the following aspects:
>
> > (3) How is the scalability of the proposed method? Although authors provided a complexity analysis of GCN-LFR, more empirical verifications on one/two large-scale graph datasets are required. My initial impression of LFR is that it may need a huge computation burden.
>
> **Preprocessing of low-frequency:** The extra complexity brought by our proposed method only lies in the acquisition of the low-frequency components. However, it only needs to be done once as a  preprocessing step.  Here, we provide the mean time of this procedure averaging from 5 runs on all datasets, including one large-scale dataset ogbn-arxiv.
>
> |                | Cora  | Citeseer | Pubmed  | Coauthor CS | Amazon Photo | ogbn-arxiv |
> |----------------|-------|----------|---------|-------------|--------------|------------|
> | LFR Burden (s) | 2.351 | 3.764    | 480.583 | 426.237     | 55.003       | 42652.283  |
>
> We find that this burden is acceptable across all datasets, especially considering that this is done on a full graph of ogbn-arxiv, which can be much faster on subgraphs if the sampling technique is used.
>
> > (3) More experimental verifications on large-scale graph benchmark datasets, for example, OGB, are required to further demonstrate the potential of the proposed method.
>
> We've tried to conduct the experiments on OGB dataset and meets two obstacles:
>
> 1. For many attack methods, scalability is even a common issue. For example, both the targeted attack method Nettack[1] and the global attack method Mettattack[2] would take weeks to finish on the full scale of ogbn-arxiv dataset. This prohibits our evaluation of large-scale OGB datasets towards robustness within a limited time.
>
> 2. Furthermore, since our theoretical analysis is based on the whole graph, we would expect to conduct the full-batch training on OGB dataset without sampling. However, we suffer a serious memory issue and can not perform the full-batch training even on the smallest datasets of OGB.  Though this could be alleviated through proper sampling, the relationship between the behavior of low-frequency components on the sampled subgraph and the whole graph is unclear and beyond the scope of our theoretical analysis.
>
> We would like to again highlight our contribution from the theoretical side. For the experimental side, we have conducted an extensive evaluation on **five** representative datasets, **three** different attacking settings and **five** popular GNN backbones. The results have demonstrated the effectiveness of our proposed models and validated our theoretical findings.
>
> [1] Adversarial Attacks on Neural Networks for Graph Data, KDD 2018
>
> [2] Adversarial Attacks on Graph Neural Networks via Meta Learning, ICLR 2019

---

> > ### Comment · Reviewer_mqEy · 2021-08-28
> > **response to authors**
> >
> > Thank you for your responses. As per my concern on OGB dataset, yes, it seems logical that your proposed method meets some obstacles. I am ok with your explanations.

---

> ### Author Response · Authors · 2021-08-10
> **The relationship of the two-channels cooperation architecture and LFR (Weakness (1-2))**
>
> We thank the reviewer for catching this vague claim through observation, and we will clarify this more clearly in the revision with the following adjustment:
>
> > (1) The reason why alternative training is better than joint training is not clearly explained.
>
> Sorry for the confusion here. Our motivation to employ alternative training is to reduce the computational cost and accelerate the training process (L266-267 in paper). Empirically,  we found the alternative training can obtain a better performance (Figure 4 in paper).
>
> > I am curious about the influence of parameter sharing used for two channels (GCN and LFR).
>
> Actually, this two-channel cooperation with parameter-sharing is a regularizing technique that aims to transfer the robustness of `LFR` to the main channel.  To further demonstrate the effectiveness of parameter-sharing, we implement a variant of GCN-LFR but only without parameter sharing, which we denote as `GCN-LFR-NoShare`. The below table demonstrates the defense performance of three models attacked by Metattack.
>
> |                            | Cora              | Citeseer          | Amazon Photo      |
> | -------------------------- | ----------------- | ----------------- | ----------------- |
> | GCN (attacked)             | 0.771 ± 0.010     | 0.654 ± 0.012     | 0.824 ± 0.004     |
> | GCN-LFR-NoShare (attacked) | 0.778 ± 0.012     | 0.663 ± 0.012     | 0.827 ± 0.014     |
> | GCN-LFR (clean/attacked)   | **0.791 ± 0.012** | **0.670 ± 0.015** | **0.843 ± 0.015** |
>
> We can observe that:
>
> 1. Even we do not share the parameters between the main channel and the LFR channel, the LFR model can enhance the defense performance.
>
> 2. The model with parameter-sharing regularization achieves the best performance. It validates the effectiveness of our architecture design.
>
> Lastly, we want to clarify that the vital factor in strengthening the robustness of models is the information extracted by `LFR`, not the parameter-sharing mechanism.

---

> > ### Comment · Reviewer_mqEy · 2021-08-28
> > **response to authors**
> >
> > Thank you for clarifying this point.

---

> ### Author Response · Authors · 2021-08-26
> **Kind reminder to the reviewer**
>
> Dear Reviewer,
>
> We are wondering whether your concerns have been properly addressed.
>
> If you have further questions after reading the answers, it would be great to let us know.
>
> Best regards,
>
> The Authors

---

> > ### Comment · Reviewer_mqEy · 2021-08-28
> > **response to authors**
> >
> > Thank you for answering my questions. Most of my concerns have been clarified. I am happy to increase my score a little bit. Hopefully, the authors can revise their work accordingly based on all the comments of the reviewers.

---

> > > ### Author Response · Authors · 2021-08-29
> > > **Thank you for your response and support**
> > >
> > > We really appreciate your time and insightful comments, and we are more than happy to find that the your concerns have been addressed.
> > >
> > > Sure, we will update the revised version accordingly as promised.
> > >
> > > If you have further concerns that you think could be clarified, we will be very glad to discuss.

---

### Decision · Program_Chairs · 2021-09-27

**Decision:**

Accept (Poster)

**Comment:**

All reviewers feel the paper is  on the borderline but leaning towards accept. The meta-reviewer read the paper and feel the insight obtained from this paper is interesting. Therefore, the meat-reviewer decides to be conformal with the rest reviewers and suggest an acceptance of this submission.